# Foldamers reveal and validate therapeutic targets associated with toxic α-synuclein self-assembly

Jemil Ahmed[1,2], Tessa C. Fitch[2,3], Courtney M. Donnelly[2,3], Johnson A. Joseph[2,3], Tyler D. Ball [2,3], Mikaela M. Bassil[2,3], Ahyun Son [2,3], Chen Zhang[4], Aurélie Ledreux[2], Scott Horowitz[1,2,3], Yan Qin [4], Daniel Paredes[2] & Sunil Kumar [1,2,3✉]

Parkinson's disease (PD) is a progressive neurodegenerative disorder for which there is no successful prevention or intervention. The pathological hallmark for PD involves the self-assembly of functional Alpha-Synuclein (αS) into non-functional amyloid structures. One of the potential therapeutic interventions against PD is the effective inhibition of αS aggregation. However, the bottleneck towards achieving this goal is the identification of αS domains/ sequences that are essential for aggregation. Using a protein mimetic approach, we have identified αS sequences-based targets that are essential for aggregation and will have significant therapeutic implications. An extensive array of in vitro, ex vivo, and in vivo assays is utilized to validate αS sequences and their structural characteristics that are essential for aggregation and propagation of PD phenotypes. The study aids in developing significant mechanistic and therapeutic insights into various facets of αS aggregation, which will pave the way for effective treatments for PD.

[1] Molecular and Cellular Biophysics Program, University of Denver, Denver, CO 80210, USA. [2] The Knoebel Institute for Healthy Aging, University of Denver, Denver, CO 80210, USA. [3] Department of Chemistry and Biochemistry, University of Denver, Denver, CO 80210, USA. [4] Department of Biological Sciences, University of Denver, Denver, CO 80210, USA. ✉email: sunil.kumar97@du.edu

Alpha-Synuclein (αS) is a neuronal protein expressed at high levels in dopaminergic neurons and it is believed to be implicated in the regulation of synaptic vesicle trafficking and recycling, and neurotransmitter release[1–8]. The misfolding of αS leads to its self-aggregation, which is a pathological hallmark of PD[1–8]. Therefore, modulation of αS aggregation is a promising therapeutic intervention for PD[1–10]. The identification and specific targeting of sequences or domains that initiate αS aggregation could promise potent antagonism of the αS self-assembly. A few small molecules have been shown to inhibit αS aggregation (Pujols et al.[11] and ref. within), however, limited atomic-level understanding is available of the ligand-αS interaction, which restricted the further optimization of the antagonists against αS aggregation. More importantly, limited progress has been made in the identification of factors that are associated with αS aggregation, e.g., αS sequences that initiate aggregation. Mutation studies enable the identification of αS sequences/domains that are important for aggregation[12–16]. However, no study has been directed to validate these αS sequences as novel targets. Here, we have utilized a foldamer-based approach in tandem with a mutation study that allowed the identification and validation of αS sequences as key therapeutic targets that are essential for the initiation of αS aggregation.

Foldamers are dynamic ligands with the ability to mimic the topography and the chemical space of the secondary structure of proteins[17–22]. The diversity of chemical space can be conveniently tuned in foldamers, an essential property for the optimization of interactions with targets. Various foldamers, including Oligoquinoline (OQ)-based and photoresponsive prion-mimics have been shown to modulate the self-assembly of islet amyloid polypeptide[22–26], Aβ peptide[21], and αS[27] whose aggregation is associated with type 2 diabetes (T2D), Alzheimer's disease (AD), and PD respectively.

We have utilized OQs to gain mechanistic and therapeutic insights into αS aggregation. Using an array of biophysical, cellular, in vivo assays and mutation studies, we have identified SK-129, a potent antagonist of αS aggregation in both in vitro and in vivo PD models. A Two-dimensional (2D) NMR-based-based atomic-level investigation enabled the identification of the binding sites of SK-129 on αS, which are validated using fluorescent polarization and mutation studies. More importantly, we have identified αS sequences as novel targets that are essential for the initiation of the aggregation. We have also validated αS sequences by targeting them with OQs and rescued PD phenotypic readouts in cellular, neuronal, and in vivo PD models. SK-129 is a potent antagonist of the αS seeds catalyzed aggregation of αS monomer. The activity of SK-129 against the αS seeds catalyzed aggregation is confirmed using distinct αS seed polymorphs generated from the recombinant αS and extracted from the substantia nigra of the post mortem brain of PD patient. The antagonist activity of SK-129 is also confirmed in a novel HEK cell-based intracellular assay for the αS seeds catalyzed aggregation of intracellular monomeric αS. Overall, SK-129 interacts at the N-terminus of αS monomer, induces or stabilizes an aggregation incompetent helical conformation, and modulates both de novo aggregation and the αS seeds catalyzed aggregation. We used a chemical tool to identify and validate αS sequences with structural insights that are essential for the aggregation and associated with PD phenotypes. The study will have significant mechanistic and therapeutic implications, which will aid in expediting treatments for PD.

## Results

**Biophysical characterization of foldamers with αS.** The OQs with carboxylic acid and hydrophobic side chains have been shown to modulate the self-assembly of amyloid proteins by specifically targeting sequences that are rich in positively charged and hydrophobic side chain residues[21–26]. Therefore, we utilized an established library of OQs with carboxylic acid and various hydrophobic groups as side chains (Fig. 1a–c). The library was screened against αS aggregation using a Thioflavin T (ThT) dye-based amyloid assay[28]. The aggregation kinetics of 100 μM αS (in 1x PBS buffer) was characterized by a sigmoidal curve with a $t_{50}$ (time to reach 50% fluorescence) of ~38.1 ± 1.8 h. The screening led to the identification of SK-129 as the most potent antagonist of wild type (WT) αS (and αS mutants, $αS_{A30P}$ and $αS_{A53T}$) aggregation at equimolar and sub-stoichiometric ratios (Fig. 1c, d and Supplementary Fig. 1). SK-129 inhibits αS aggregation under both de novo and lipid membrane conditions at an equimolar ratio (Supplementary Fig. 2a), which was also validated by transmission electron microscopy (TEM) images in the absence (Supplementary Fig. 2b) and presence of SK-129 (Supplementary Fig. 2c). Under matched conditions, we did not observe any significant quenching of the ThT fluorescence signal by SK-129 (Supplementary Fig. 3). The antagonist activity of SK-129 for αS aggregation was also analyzed using SDS-PAGE (Supplementary Fig. 4). A solution of 100 μM αS was aggregated for four days in the aggregation buffer (20 mM NaCl, 20 mM NaPi, pH 6.5) in the absence and presence of SK-129 at an equimolar ratio. Subsequently, the αS solutions were centrifuged to separate αS aggregates from the soluble αS. Afterward, the samples were boiled at 95 °C for 5 min. to disassemble αS aggregates and examined them using sodium dodecyl sulfate–polyacrylamide gel electrophoresis (SDS-PAGE) (Supplementary Fig. 4). In addition, we also quantified SDS-PAGE gel band intensities using ImageJ software (Supplementary Fig. 4b, c). In the absence of SK-129, αS was predominantly detected in the insoluble fraction (αS aggregates) (Supplementary Fig. 4a, b). In marked contrast, in the presence of SK-129, αS was predominantly detected in the soluble fraction (αS monomer) (Supplementary Fig. 4a, c). These results clearly demonstrate that SK-129 is a potent inhibitor of αS aggregation.

The antagonist activity of SK-129 is predominantly a consequence of the side chains. Among analogs (with varying hydrophobicity), SK-129 was the most potent antagonist of αS aggregation (Fig. 1e, f and Supplementary Fig. 5), which indicates that moderate hydrophobicity at positions 1 and 3 is required to achieve the optimal activity. The positioning of the side chains was important for SK-129's activity as scrambling of side chains led to significantly diminished activity (Fig. 1f and Supplementary Figs. 6 and 7). The antagonist activity of SK-129 was much better than Epigallocatechin Gallate (EGCG), a potent antagonist of αS aggregation[29]. Gel shift assay shows that SK-129 potently inhibits αS aggregation; however, higher-order toxic oligomers ($n > 5$)[30,31] were observed in the presence of EGCG (Supplementary Fig. 8). We next determine the binding affinity of SK-129 using fluorescence polarization (FP) titration between a fluorescent analog of SK-129 ($SK-129_F$) (Supplementary Fig. 5) and αS, which yielded a $K_d$ of 0.81 ± 0.09 μM (Fig. 1g) and a binding stoichiometry of 1:1 ($αS:SK-129_F$) (Supplementary Fig. 9). The complex of $SK-129_F$-αS was used for a displacement titration with SK-129, which yielded a $K_d$ of 0.72 ± 0.06 μM (Supplementary Fig. 10). More importantly, $SK-129_F$ ($SK-129_F$-αS complex) could be used as a novel tool for a high throughput assay to screen and identify high-affinity ligands for αS. Also, SK-129 is a very specific antagonist of αS aggregation. The $K_d$ of SK-129 for $Aβ_{42}$ was more than 15-fold higher than αS (Supplementary Fig. 11a). Corroborating this specificity, SK-129 did not show any noticeable effect on Aβ aggregation (Supplementary Fig. 11b).

The aggregation of αS is associated with toxicity;[1–8] therefore, we tested the efficacy of SK-129 on the neuroblastoma SH-SY5Y cell line. The aggregation of αS was carried out in the absence and presence of SK-129 at an equimolar ratio, and the solutions were tested in SH-SY5Y[32]. The viability of SH-SY5Y cells decreased to 48 ± 3% upon

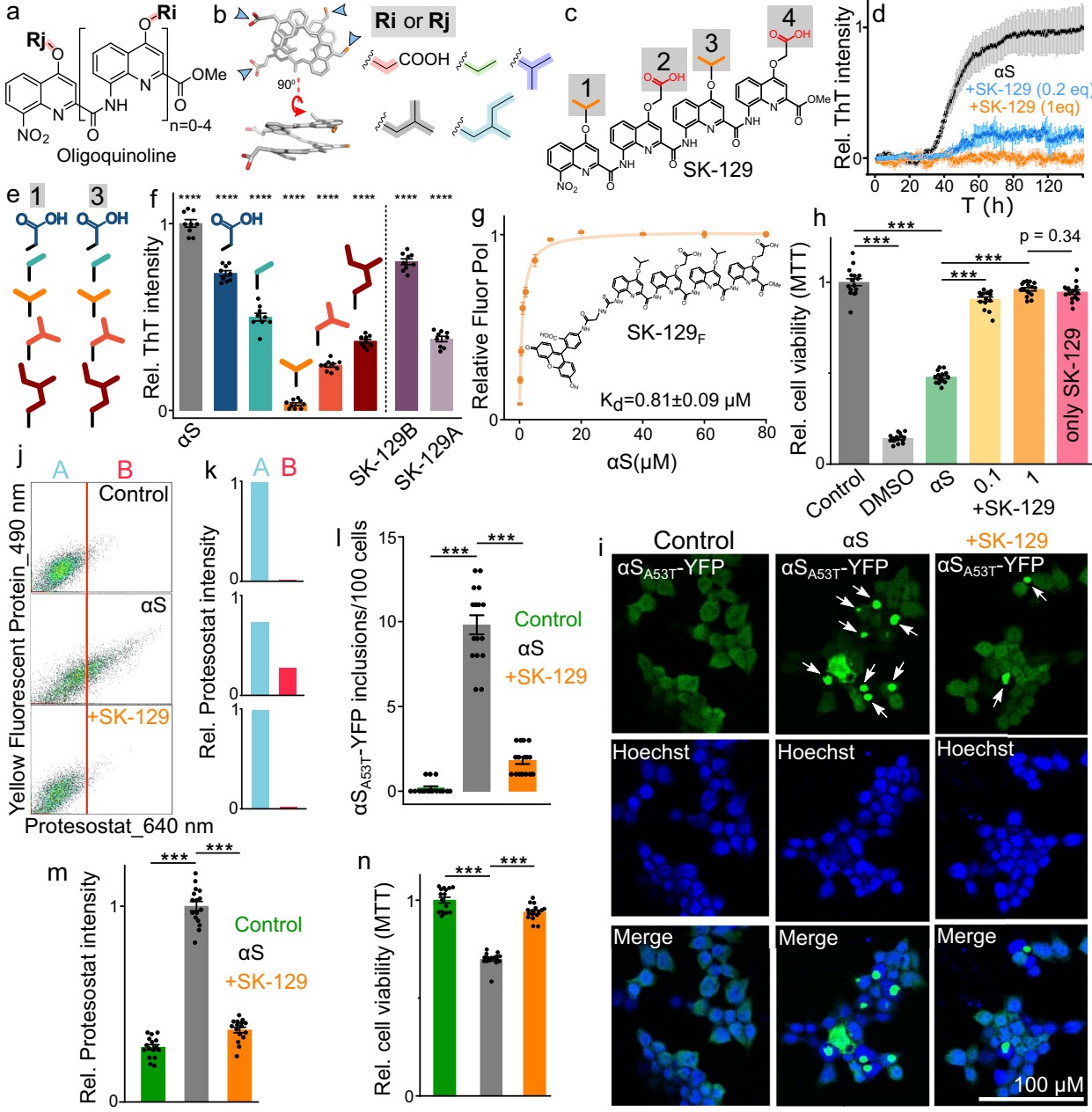

exposure to 10 μM αS for 24 h; which was rescued to 91 ± 4% and 97 ± 4% in the presence of SK-129 at molar ratios of 1:0.1 and 1:1 (αS:SK-129), respectively (Fig. 1h). The rescue of toxicity by SK-129 in SH-SY5Y cells was similar at higher concentrations of αS (25 μM and 50 μM) (Supplementary Fig. 12). To confirm that SK-129 did not generate seed-competent structures during the inhibition of αS aggregation, we utilized two HEK cell lines, which stably express YFP-labeled WT αS (αS-YFP) and a familial mutant, A53T (αS-$_{A53T}$-YFP)[33,34]. Both HEK cells have been shown to template endogenous monomeric αS-$_{A53T}$-YFP (αS-YFP) into fibers when transfected with αS fibers with lipofectamine 3000 (Fig. 1i)[33,34], which is detected by intracellular fluorescent puncta (Fig. 1i, white arrows). αS was aggregated in the absence and presence of SK-129 at an equimolar ratio and the HEK cells were treated with αS fibers (7 μM in monomeric αS) for 24 h (Fig. 1i). A very small number of inclusions (αS-$_{A53T}$-YFP or αS-YFP) were observed in the presence of SK-129 (Fig. 1i). The inclusions (αS-$_{A53T}$-YFP) were quantified by confocal microscopy (Fig. 1i, l), flow cytometry (intracellular inclusions stained with ProteoStat dye[35], Fig. 1j, k), and a 96-well plate reader

(using ProteoStat dye, Fig. 1m), which were alleviated significantly in the presence of SK-129. The viability of HEK cells improved from 68% to 94% (αS aggregated solution in the presence of SK-129), which was determined using the (3-(4,5-dimethylthiazol-2-yl)-2,5-diphenyltetrazolium bromide) (MTT) reduction-based cytotoxicity assay (Fig. 1n). We used all the above-mentioned techniques to assess the antagonist activity of SK-129 on the aggregation of WT αS in HEK cells as well (Supplementary Fig. 13). SK-129 was equally effective in inhibiting aggregation and cytotoxicity in HEK cells that were expressing intracellular αS-YFP, which was confirmed with confocal imaging (Supplementary Fig. 13a), flow cytometry (Supplementary Fig. 13b–d), ProteoStat dye-based quantification of intracellular inclusions (Supplementary Fig. 13e, f), and MTT based cytotoxicity assay (Supplementary Fig. 13g).

**Identification of the binding site of SK-129 on αS.** The N-terminal domain spanning residues 1–90 plays a significant role in αS aggregation[8,12,36–40]. Therefore, we hypothesized that

**Fig. 1 Characterization of the antagonist activity of SK-129 against αS aggregation. a** The generic chemical structure of the OQ with Ri and Rj are the side chain surface functionalities. The side and top view of the crystal structure of OQs and the surface functionalities are represented by arrows. The OQs with the indicated side chains (Ri and Rj) were used in the study. **c** Chemical structure of SK-129 and the four side chains were indicated from 1 to 4. **d** The average of ThT-dye fluorescence-based aggregation profile of 100 μM αS in the absence and presence of SK-129 at the indicated molar ratios. The data were expressed as mean and the error bars report the S.D. ($n = 3$ independent experiments). **e** The chemical structures of the side chains at position 1 and 3 of various analogs of SK-129. **f** The antagonist activities of the analogs (100 μM) of SK-129 against 100 μM αS aggregation. The data were expressed as mean and the error bars report the s.e.m. ($n = 3$ independent experiments and each n consisted of 3 technical replicates). **g** The fit for the FP titration curve to determine the binding affinity between 10 μM SK-129$_F$ and αS. The chemical structure of SK-129$_F$ is shown as well. The data were expressed as mean and the error bars report the s.d. ($n = 3$ independent experiments). **h** The statistical analysis of the relative viability of SH-SY5Y cells when treated with the aggregated solution of 10 μM αS in the absence and presence of SK-129 at the indicated molar ratios. The data were expressed as mean and the error bars report the s.e.m. ($n = 4$ independent cell toxicity experiments and each n consisted of 4 technical replicates). **i** Confocal images of HEK cells treated with the aggregated solution of 7 μM αS in the absence and presence of SK-129 at an equimolar ratio. Inclusions of αS$_{A53T}$-YFP = white arrows, Hoechst (blue), merge = Hoechst and αS$_{A53T}$-YFP. **j** The flow cytometry-based analysis of HEK cells treated with the aggregated solution of 7 μM αS in the absence and presence of SK-129 at an equimolar ratio. The x-axis represents αS$_{A53T}$-YFP aggregates containing cells that are stained with Proteostat dye ($\lambda = 640$ nm) and the y-axis represents the total number of cells with YFP ($\lambda = 490$ nm). **k** Columns A and B represent the relative % of HEK cells without and with αS$_{A53T}$-YFP aggregates, respectively. **l** The number of αS$_{A53T}$-YFP inclusions when HEK cells were treated with the aggregated solution of 7 μM αS in the absence and presence of SK-129 at an equimolar ratio. A total of 100 HEK cells were examined to count the number of inclusions at four different locations in the eight-well plate for each experiment and it was repeated in four independent experiments. The relative intensity of Proteostat dye-stained aggregates of αS$_{A53T}$-YFP inclusions (**m**) and relative viability (**n**) of HEK cells treated with the aggregated solution of 7 μM αS in the absence and presence of SK-129 at an equimolar ratio. The data (for **l–n**) were expressed as mean and the error bars report the s.e.m. ($n = 4$ independent HEK cells-based experiments and each $n$ consisted of four technical replicates). The statistical analysis was performed using one-way analysis of variance (ANOVA) with Tukey's multiple comparison test. *$p < 0.05$, **$p < 0.01$, ***$p < 0.001$. Source data are provided as a Source Data file.

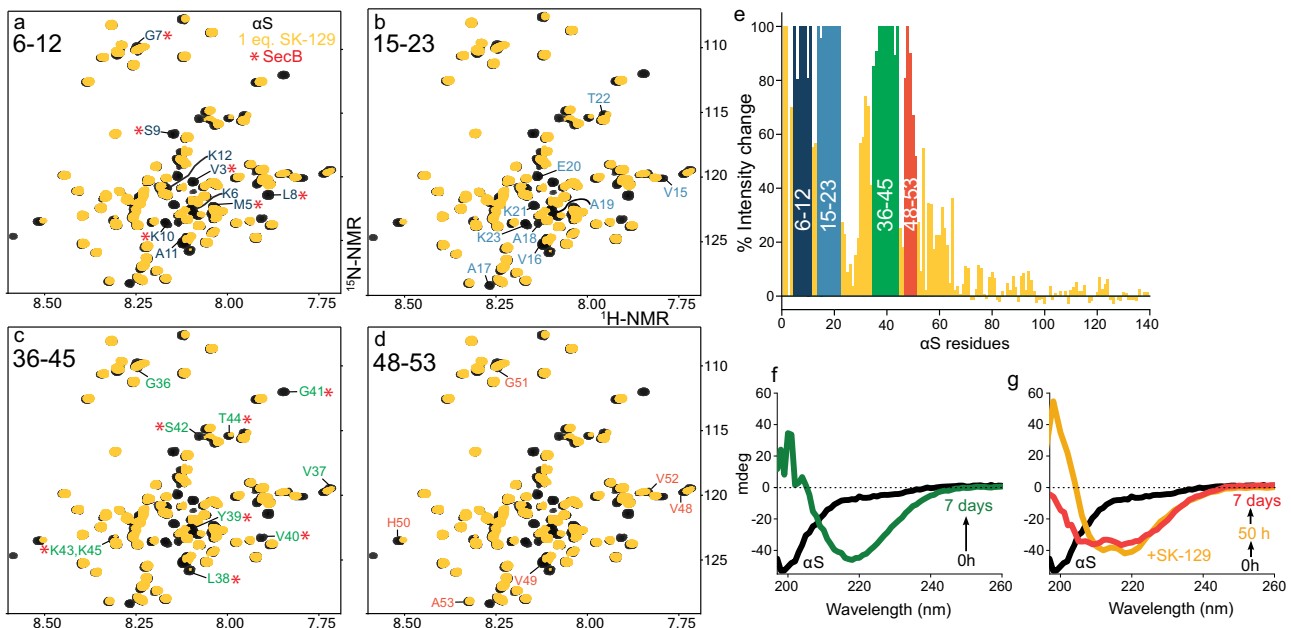

**Fig. 2 Structural characterization of the binding interaction between SK-129 and αS. a–d** Overlay of 2D HSQC ($^1$H, $^{15}$N) NMR spectra of 70 μM uniformly $^{15}$N-labeled αS in the absence (black) and presence (yellow) of SK-129 at an equimolar ratio. The largest attenuation in the volume of the backbone amide residue NMR signals are highlighted and assigned, which includes αS segments from 6–12 (**a**), 15–23 (**b**), 36–45 (**c**), and 48–53 (**d**). The change in the volume of amide backbone residue peaks of αS was compared between SK-129 and a molecular chaperone SecB (*, red) and the pronounced changes were observed in segments 6–12 (**a**) and 36–45 (**b**). **e** Graphical presentation of the changes in the chemical shifts of the backbone amide residue peaks of $^{15}$N-labeled αS (70 μM) in the presence of SK-129 at an equimolar ratio. The colored sequences are the potential binding sites of SK-129 on αS. CD-based characterization of the aggregation kinetics of 35 μM αS in the absence (**f**) and presence (**g**) of SK-129 at an equimolar ratio. The spectra were recorded for 7 days. Source data are provided as a Source Data file.

SK-129 could be interacting with the N-terminus of αS for the potent inhibition of αS aggregation. We utilized 2D heteronuclear single quantum coherence NMR spectroscopy (2D HSQC NMR) for atomic-level insight into the binding site of SK-129 on αS. We collected the HSQC NMR of 70 μM $^{15}$N-$^1$H-uniformly labeled αS in the absence and presence of SK-129 at an equimolar ratio (Fig. 2a–e) and compared the signal intensity of the amide peaks. The total changes in the intensity in the presence of SK-129

suggest that the binding site of SK-129 is toward the N-terminus of αS (Fig. 2a–e), more specifically SK-129 interacts and changes the conformation of four αS sequences, including 6–12, 15–23, 36–45, and 48–53 (Fig. 2a–e). The binding sites of SK-129 on αS contain lysine and hydrophobic residues; therefore, we propose that the carboxylic acid and the propyl side chains of SK-129 are involved in binding interactions with lysine and hydrophobic residues of αS.

**Effect of SK-129 on αS conformation**. The NMR study also suggests that SK-129 induces α-helical conformation in αS. The intensity changes of αS residues in the presence of large uni-lamellar vesicles [0.875 mM, LUVs, 100 nm, DOPS, 1,2-dioleoyl-sn-glycero-3-phospho-L-serine (sodium salt)] were similar to those influenced by SK-129 at a higher molar ratio (1:2, αS:SK-129) (Supplementary Figs. 14–16). As αS samples α-helical con-formations in the presence of LUVs[36–40], we postulated that αS forms an α-helical conformation in the presence of SK-129. We utilized circular dichroism (CD) to study the interaction of αS with SK-129. The conformation of 35 μM αS transitioned from random coil to β-sheet in 7 days (Fig. 2f); however, the con-formation of αS switched from random coil to α-helix and stayed in the same conformation in the presence of SK-129 at an equimolar ratio (Fig. 2g). We posit that the antagonist activity of SK-129 against αS aggregation is a consequence of the direct interaction with the N-terminus and the induction or stabilization of an α-helical conformation in αS. SK-129 behaved similarly under lipid catalyzed αS aggregation. SK-129 was a potent antagonist of LUVs catalyzed αS aggregation (Supplementary Fig. 2a). The CD spectra of 30 μM αS switched from an α-helix to a β-sheet conformation via an α-helical conformation in the presence of LUVs (375 μM, 100 nm, DOPS, Supplementary Fig. 17a)[36–43]. In marked contrast, αS remained in an α-helix conformation in the presence of LUVs and SK-129 at an equi-molar ratio (Supplementary Fig. 17b). However, there was a decrease in the CD intensity of α-helix upon the addition of SK-129, which suggests that SK-129 might be competing against the lipid membrane for αS (Supplementary Fig. 17b). We utilized a CD study to determine the $K_d$ between αS and LUVs (DOPS, 100 nm), similar to previously published work[38,42]. A CD titra-tion was carried out between 40 μM αS and an increasing conc. of DOPS (molar ratio, 1:160, αS:DOPS), which yielded a $K_d > 5$ μM ($K_d = 7.2 ± 2.5$ μM, Supplementary Fig. 18). The $K_d$ between αS and DOPS was higher than the $K_d$ between SK-129 and αS ($0.72 ± 0.06$ μM), which indicates that the binding affinity of SK-129 is higher than DOPS for αS. Consequently, αS should favor binding to SK-129 than DOPS, when SK-129 is added to the complex of αS + DOPS. Our CD data support this claim as the addition of SK-129 to the αS + DOPS complex resulted in a decrease in the CD signal intensity of the DOPS bound αS. Under these conditions, SK-129 does not form any micelle structures, as confirmed with the TEM images (Supplementary Fig. 19). Overall, the data suggest that SK-129 specifically interacts with αS and competes with the LUVs for αS and inhibit the aggregation of lipid catalyzed aggregation.

Collectively, our data from CD and NMR suggest that the conformation of αS remains in the α-helical state in the presence of SK-129 for the whole time course of the experiments. The data also indicate that αS was not completely displaced from lipid membranes in the presence of SK-129 and the lipid catalyzed aggregation of αS was wholly inhibited by SK-129. Our CD and NMR data suggest that the inhibition of the membrane-catalyzed aggregation of αS might be a consequence of the competition of αS between lipid membranes and SK-129. A recent study has suggested that one of the main therapeutic strategies could be the inhibition of αS aggregation on lipid membranes without completely displacing αS from lipid membranes[42,43]. The native function of αS is partly facilitated by its interaction with lipid membranes and the complete displacement of αS from lipid membranes could be detrimental to its function and might promote neuropathology[42,43]. A natural product, Squalamine, was able to inhibit the membrane-potentiated αS aggregation and rescued cytotoxicity by comple-tely displacing αS from lipid membranes[42,43]. However, SK-129 was able to inhibit the aggregation of αS without completely

displacing it from lipid membranes, which is evidenced by the intact α-helical conformation of αS in the presence of SK-129 under lipid membrane conditions.

We also employed HSQC 2D-NMR to gain molecular insights into the mode of action of SK-129 on lipid membrane catalyzed aggregation of αS and the overall effect of SK-129 on the membrane bound-αS complex. SK-129 (140 μM) was added to the complex of [15]N αS:LUVs (70 μM: 875 μM) and the intensity changes of the amide peaks from this NMR (Supplementary Fig. 20) were compared with the amide peaks of the NMR from the αS:SK-129 complex (Supplementary Figs. 14 and 16) and the αS:LUVs complex (Supplementary Figs. 15 and 16). The addition of SK-129 to the αS-LUVs complex leads to the dissapperance of various amide peaks in the NMR spectrum (Supplementary Fig. 20). If SK-129 was able to completely displace αS from the LUVs, this NMR spectrum should have been similar to the NMR spectrum of the αS-SK-129 complex (Supplementary Fig. 14). However, the NMR spectrum was not similar to that of the αS-SK-129 complex or the αS-LUVs complex. The NMR spectrum was a combination of the NMRs of αS-SK-129 and αS-LUVs complexes, which suggests an interchange of αS between SK-129 and LUVs (Supplementary Fig. 20). The NMR also suggest that SK-129 did not completely displaced αS from the LUVs. Our study demonstrates that SK-129 was able to inhibit membrane catalyzed αS aggregation without completely displacing αS from lipid membranes. The study suggesting that SK-129 is likely not interfering with the native function of αS, which is partly facilitated by the interaction of αS with the lipid membranes.

To further confirm the binding sites of SK-129 on αS, we carried out a mutation study by systematically removing residues 6-12, 15-23, 36-45, or 48-53 from WT αS denoted as αS1, αS2, αS3, and αS4, respectively (Fig. 3a). The mutants were expressed and characterized using gel shift assay (Supplementary Fig. 21) and mass spectrometry (Supplementary Fig. 22a–f). The FP-based binding affinity of SK-129F for αS1 and αS2 mutants was 3–4-fold weaker (than WT αS) and very weak for both αS3 (~8 fold) and αS4 mutants (>10 fold, Fig. 3b–e). We posit that SK-129 has multiple binding sites on αS with varying binding affinities or that the main binding site spans residues 36–53, and the intensity change of residues 6–12 and 15–23 is a consequence of the conformational switch in αS.

**αS sequences essential for de novo and seed catalyzed aggre-gation**. SK-129 inhibits aggregation by interacting with four αS sequences; therefore, we hypothesize that these sequences might be essential to initiate αS aggregation. Therefore, we investigated the effect of these sequences on αS aggregation. Mutants αS3 and αS4 did not aggregate under our conditions via ThT and TEM (Fig. 3a, i, j, n, o and Supplementary Fig. 23), and their CD spectra were random coil (Fig. 3s, t). Mutants αS1 and αS2 aggregated with $t_{50}$'s 3–4-fold higher than WT αS (Fig. 3a, f, g and Supplementary Fig. 23). The morphology of αS1 fibers was similar to WT αS (Fig. 3l); however, αS2 fibers were amorphous (Fig. 3l). Both WT αS and αS1 sampled β-sheet conformation (Fig. 3p, q); however, αS2 did not have the characteristics of a β-sheet conformation (Fig. 3r).

We also investigated the role of these αS sequences on the seed-catalyzed aggregation of αS. The WT αS seeds (10% monomer concentration) accelerated 100 μM αS aggregation by decreasing the $t_{50}$ of WT αS ($28.3 ± 2.2$ h), αS1 ($84.1 ± 3.6$ h), αS2 ($72.1 ± 3.4$ h) to WT αS ($8.9 ± 0.2$ h), αS1 ($14.0 ± 0.6$ h), αS2 ($21.6 ± 0.7$ h) (Fig. 3u–y and Supplementary Fig. 24). The αS seeds did not template and aggregated mutants αS3 and αS4 (Fig. 3x, y and Supplementary Fig. 24), which suggests that the deleted sequences in αS3 and αS4 might be involved in seed

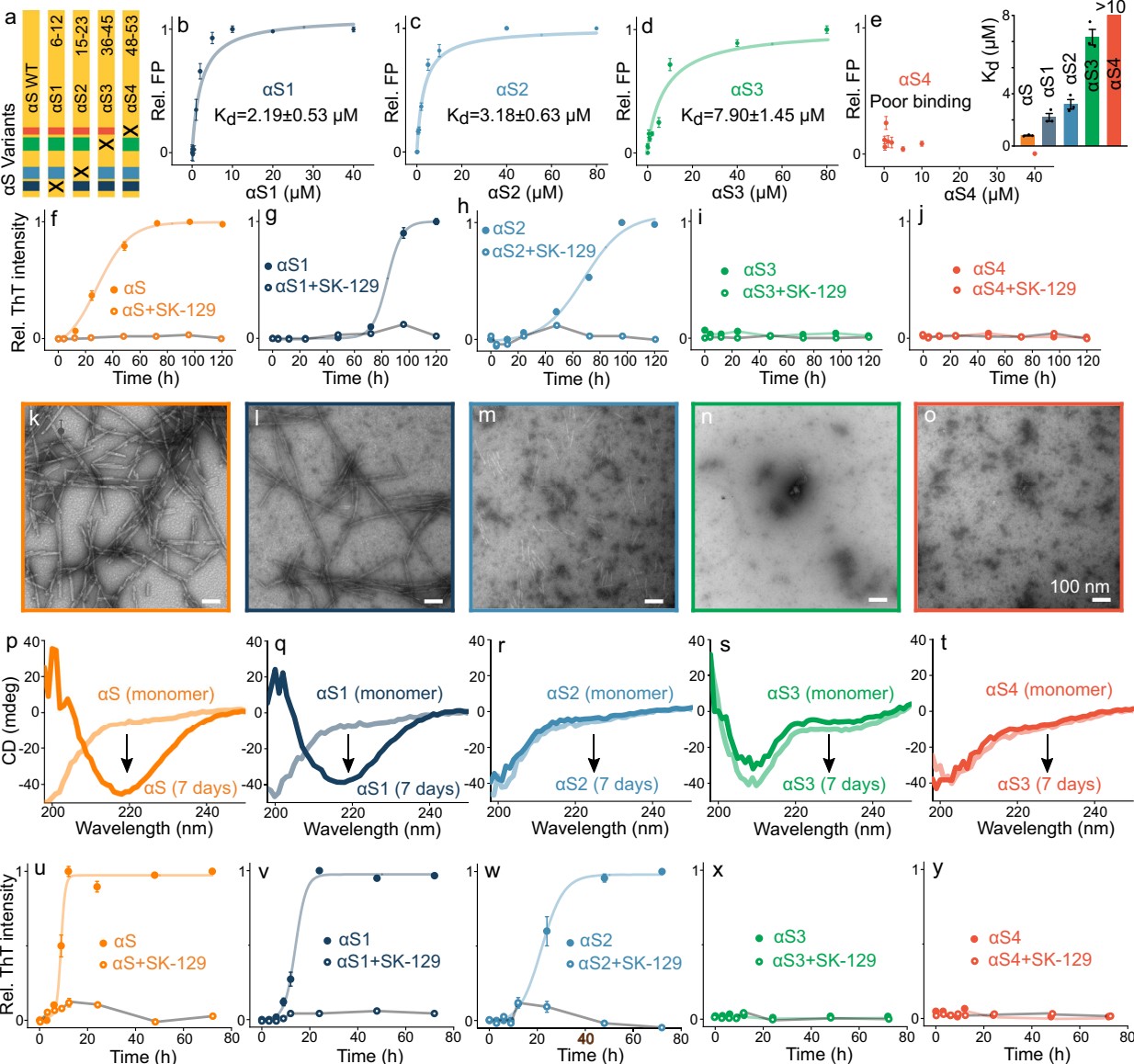

**Fig. 3 SK-129-based mutation study to delineate the role of αS sequences in aggregation. a** A schematic of the design of αS variants where "x" represents the deleted sequence from the WT αS. **b–e** The fits for the FP titrations to determine the binding affinities between 10 μM SK-129$_F$ and αS variants (inset). The data were expressed as mean and the error bars report the S.D. ($n = 3$ independent experiments). **f–j** ThT fluorescence-based aggregation kinetic profiles of 100 μM αS variants in the absence (closed circle) and presence (open circle) of SK-129 at an equimolar ratio. The circles represent the average ThT intensity of three different experiments. The data were expressed as mean and the error bars report the S.D. ($n = 3$ independent experiments). **k–o** TEM images of αS variants (100 μM) after aggregated them for seven days. **p–t** CD spectra of monomeric (light color, monomer) and the aggregated (dark color, 7 days) states of αS variants (35 μM). The same aggregated samples of αS variants were used for both CD and TEM images. **u–y** Aggregation profiles of 100 μM αS variants catalyzed by preformed fibers of WT αS (10 μM in monomeric unit) in the absence (close circle) and presence (open circle) of SK-129 at an equimolar ratio. The data were expressed as mean and the error bars report the S.D. ($n = 3$ independent experiments). Source data are provided as a Source Data file.

catalyzed aggregation. SK-129 wholly suppressed the seed-catalyzed aggregation of WT αS, αS1 and αS2 (Fig. 3u–w and Supplementary Fig. 24) at an equimolar ratio. These experiments show that the sequences affected by SK-129 are important for αS aggregation.

The antagonist activity of SK-129 on αS aggregation was also assessed using a protein misfolding cyclic amplification (PMCA) technique. The PMCA technique is used to cyclically amplify the aggregation of proteins from a small quantity and diverse species and it also generates robust seeds via a nucleation-dependent polymerization model[44–46]. In the PMCA assay, αS fibers are

amplified for five cycles using αS monomer and seeds from the previous cycle. Additionally, αS seed polymorphs from different sources differ in mediating PD phenotypes, which is a consequence of their spread through various infection pathways[47–49]. Therefore, we utilized two αS seed polymorphs, including recombinant αS seeds and αS seeds extracted from the substantia nigra of a PD brain and a control brain (post mortem condition) and used them in the PMCA assay (Fig. 4a). The Lewy bodies (LBs)-like structural features in the substantia nigra of the PD brain were confirmed using immunostaining (bluish/black, black arrows) (Fig. 4c). We did not observe any LBs-like

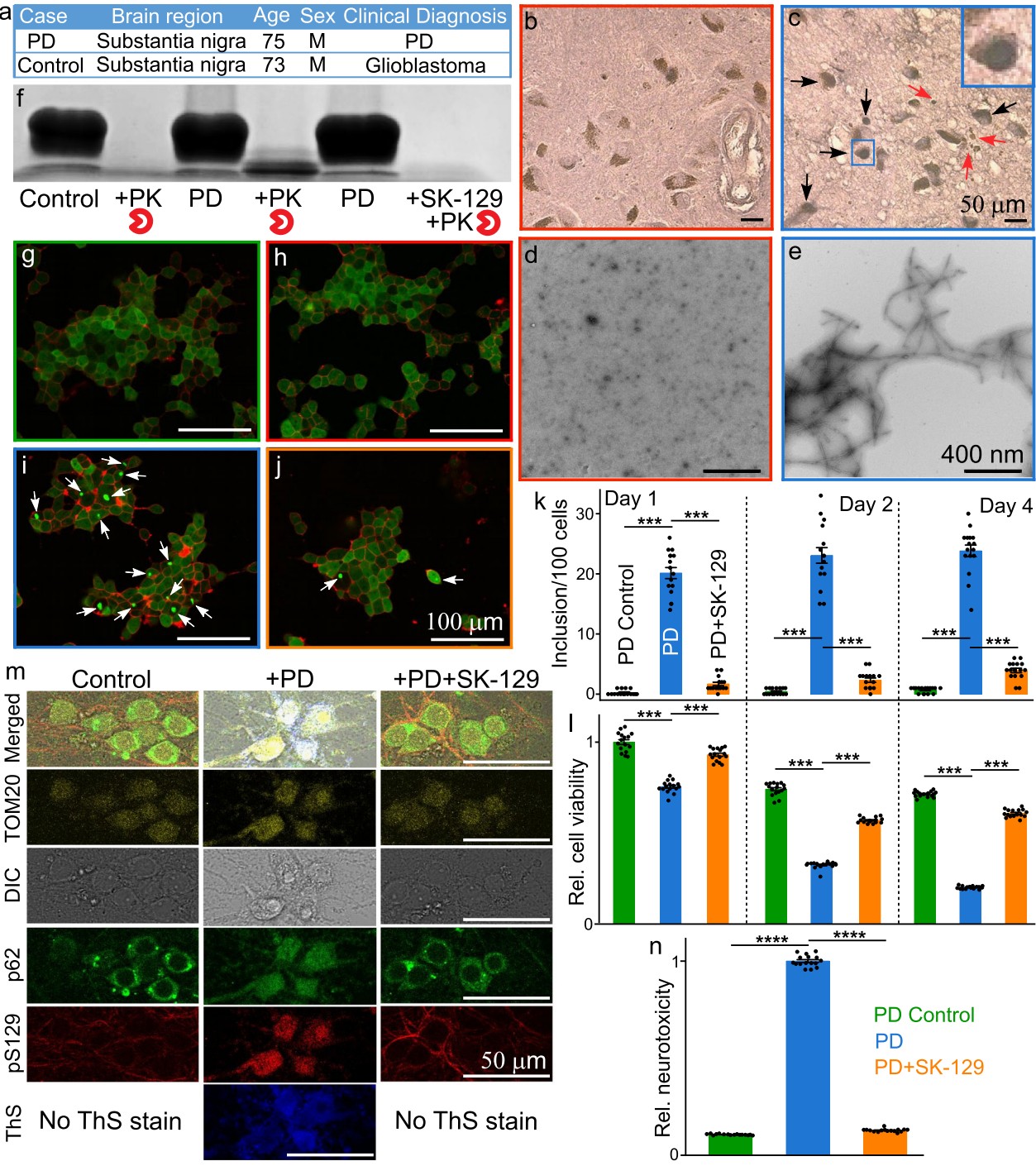

structural features in the control brain (Fig. 4b). The aggregates of αS were extracted from the PD brain using a published protocol[34]. The αS aggregates were extracted and confirmed from the PD brain using TEM (Fig. 4e) and western blot (Fig. 4f). In marked contrast, no aggregates of αS were detected after extraction from the control brain as confirmed by TEM (Fig. 4d). Both samples from the control brain and the PD brain were assessed for their ability to seed and accelerate the aggregation of αS monomer. No noticeable change was observed in the $t_{50}$ or the total ThT intensity for αS agregation in the presence of the control brain sample. The $t_{50}$ for αS aggregation was 67.4 ± 12.2 h and 62.1 ± 3.4 h in the absence and presence of the control brain sample, respectively (Supplementary Fig. 25a, b). In marked

contrast, both $t_{50}$ (22.7 ± 5.4 h) and ThT intensity (~4 fold) for αS agregation were significantly enhanced in the presence of the PD brain sample (Supplementary Fig. 25a, b). Clearly, the αS seeds from the PD brain sample template and significantly accelerate αS aggregation via seed catalyzed mechanism. The PMCA assay sample (cycle 5) of the control brain extract was not PK (proteinase K) resistant (Fig. 4f) and it was also not effective at templating αS-$_{A53T}$-YFP monomer into inclusions in HEK cells (Fig. 4h). However, the sample (cycle 5) from the PD brain was PK resistant (Fig. 4f) and it was also effective at templating αS-$_{A53T}$-YFP monomer into inclusions in HEK cells (Fig. 4i, k). The inclusions and toxicity increased gradually up to four days in the presence of the seeds from PD brain sample (Fig. 4i, k, l).

**Fig. 4 The assessment of the antagonist activity of SK-129 in ex vivo PD models. a** The demographic and clinical information of the human brain tissues. Neuromelanin (brown) and αS immunostaining (LBs-like structure, bluish/black, black arrows, inset) in substantia nigra neurons from control (**b**) and PD (**c**) post mortem brain. Degenerating neurons and the extracellular neuromelanin debris from dying neurons (red arrows) were also visible. The hollow spaces in the PD brain demarcate cell loss. (Inset) A zoom in view of LB-like structure. TEM images of the αS seeds extracted from the control (**d**) and PD brains (**e**). **f** The αS stained western blot of the PMCA sample from the fifth cycle of the control and PD brain extracts after treatment with PK in the absence and presence of SK-129. Confocal images of HEK cells after treatment with control (**g**), control (**h**), and PD (**i**) brain extracts from PMCA sample (fifth cycle) and in the presence of SK-129 (**j**) at an equimolar ratio (**j**). The number of inclusions (**k**) and relative viability (**l**) of HEK cells in the presence of PMCA samples (fifth cycle) from PD brain extracts under the indicated conditions. A total of 100 HEK cells were examined to count the number of inclusions at four different locations in the eight-well plate for each experiment and it was repeated in four independent experiments. The data (for **k**, **l**) were expressed as mean and the error bars report the s.e.m. ($n = 4$ independent HEK cells-based experiments and each n consisted of 4 technical replicates). **m** Confocal imaging of primary neurons treated with PMCA samples (fifth cycle) of control and PD brain extracts in the absence and presence of SK-129 at an equimolar ratio for 21 days. The primary neurons were stained with various markers, including LB biomarkers (pS129 and p62), mitochondria marker (TOM20), and aggregate staining ThS dye. **n** Under matched conditions (to **m**), the neurotoxicity of primary neurons was measured using the LDH assay. The data were expressed as mean and the error bars report the s.e.m. ($n = 4$ independent LDH experiments and each n consisted of four technical replicates). The data were expressed as mean and the error bars report the s.e.m. ($n = 4$ independent experiments and each $n$ consisted of four technical replicates). The statistical analysis was performed using ANOVA with Tukey's multiple comparison test. *$p < 0.05$, **$p < 0.01$, ***$p < 0.001$, ****$p < 0.0001$. Source data are provided as a Source Data file.

Under the biological condition used, the control HEK cells were healthy up to 4 days and therefore, we decide to restrict our study up to 4 days.

**Effect of SK-129 on the seed catalyzed aggregation of αS in an ex vivo PD model.** We next investigated the effect of SK-129 at preventing seed-catalyzed aggregation from PD brain samples. The sample (cycle 5) of PD brain extract in the presence of SK-129 was neither aggregated nor PK resistant (Fig. 4a). Also, we observed a lower number of inclusions and improved cell viability for up to four days (Fig. 4j–l). We observed similar behavior of the PMCA sample (cycle 5) from recombinant αS seeds in the absence and presence of SK-129 at an equimolar ratio (Supplementary Fig. 26). The PMCA assay for recombinant αS leads to an abundance of αS fibers (Cycle 5), confirmed with high ThT signal (Supplementary Fig. 26f), TEM image (Supplementary Fig. 26d), PK resistance (Supplementary Fig. 26b, white arrows), high number of inclusions from confocal imaging (Supplementary Fig. 26g–i), high ProteoStat dye signal (Supplementary Fig. 26j), and much higher cytotoxicity (Supplementary Fig. 26k) in HEK cells. In contrast, there was no formation of αS fibers for the PMCA assay (Cycle 5) in the presence of SK-129 at an equimolar ratio as confirmed by low ThT intensity (Supplementary Fig. 26f), TEM image (Supplementary Fig. 26e), no PK resistance (Supplementary Fig. 26c, orange arrows), very low number of inclusions from confocal imaging (Supplementary Fig. 26g, i), low ProteoStat dye signal (Supplementary Fig. 26j) and rescue of cytotoxicity in HEK cells (Supplementary Fig. 26k).

To further confirm the antagonist activity of SK-129 on the seed catalyzed aggregation of αS in the presence of the PD brain extract, we employed a more physiologically relevant model based on the primary rat hippocampal neurons[50,51]. Using primary hippocampal neurons, an αS aggregation-based seeding model has been recently developed that recapitulates the key events of aggregation, seeding, and maturation of inclusions that partly mimic the features of LB-like structures[50]. We incubated primary culture neurons for a total of 31 days, including a 10 day of incubation period, followed by the addition of PMCA samples (+PD or PD + SK-129) and incubation for another 21 days. The reported total incubation time (in literature) for the primary culture neurons was much shorter (maximum time = 21 days)[50,51]; however, we incubated the primary culture neurons for a total of 31 days. The reason for the longer incubation time for the primary neurons in the presence of PD sample was because we did not observe any significant

intracellular aggregation and neurotoxicity at shorter incubation times in the presence of the PD sample. The difference in the incubation time (literature vs our experiment) required to induce neurotoxicity in the primary culture neurons was likely due to the difference in the αS fibril polymorphs of our experiment and the literature sample. It has been shown earlier that different αS fibril polymorphs could differ in templating αS aggregation and inducing toxicity[47–49]. At 31 days of incubation time, we observed both aggregation and significant neurotoxicity in the primary culture neurons in the presence of PD fibrils. The primary culture neurons treated with the PD sample (Fig. 4m) were stained after 31 days and they were stained positive for αS-pS-129 (phosphorylated residue 129 in WT αS) (red color) and ThS (blue color), a dye that specifically binds protein aggregates (Fig. 4m)[52]. In addition, these aggregates were stained positive and colocalized for p62 (autophagosome vesicles) and TOM20 (mitochondria) as well (Fig. 4m) and the aggregates were most likely colocalized in the cytoplasmic region of the neurons as suggested by others as well[50,51]. Our confocal imaging data corroborate well with the earlier published work[50,51]. The data suggest that αS inclusions recruit and sequester various organelles, proteins, and membranous structures, similar to the published work[50,51,53,54]. The staining profile of the primary neurons was very similar for both the control and PD fibrils+SK-129 conditions. We did not observe any colocalization of αS-pS-129 with p62, TOM20 in both the control and PD + SK-129 conditions (Fig. 4m, +PD + SK-129). Also, no staining of αS-pS-129 was observed with ThS dye for both the control and PD + SK-129 conditions (Fig. 4m). We observed mild staining and diffusion of αS-pS-129 in both the control and PD + SK-129 conditions; however, we did not observe any colocalization of αS-pS-129 with any other biomarker, including ThS dye (Fig. 4m). The partial staining of αS-pS-129 is likely due to the longer incubation time (31 days) for the primary culture neurons in our experimental conditions, which might have contributed to some neurotoxicity and the mild staining of αS-pS-129. We used lactate dehydrogenase (LDH) release assay to determine the neurotoxicity of the primary culture neurons in the presence of PD fibrils (±SK-129). We observed very high neurotoxicity (~8-fold higher than control) in primary neurons in the presence of the PD sample. In marked contrast, similar to the control sample, we did not observe any significant neurotoxicity in the presence of PD + SK-129 condition (Fig. 4n).

To further validate these results, we used HEK cells-based model, which expresses endogenous monomeric αS-$_{A53T}$-YFP. In the presence of PD fibrils, both P62 and αS-pS-129 colocalized in

the aggresome of αS inclusions in HEK cells after 24 h (Supplementary Fig. 27). The αS inclusions were colocalized in the cytoplasmic region of the HEK cells as suggested by others as well[51]. Our results corroborate well with the earlier published work with HEK cells (Supplementary Fig. 27a, b)[51]. In marked contrast, in the presence of PD fibrils+SK-129 condition, we detected a significantly smaller number of colocalization of P62 and αS-pS-129 in the aggresome of αS inclusions (Supplementary Fig. 27a, b). In addition, we carried out the MTT reduction-based cytotoxicity assay for the HEK cells in the presence of PD fibrils. The cell viability of HEK cells decreased to 50.5 ± 5.4% in the presence of PD fibrils (Supplementary Fig. 27c). However, in the presence of PD fibrils+SK-129 condition, the cell viability increased to 85.6 ± 7.8% (Supplementary Fig. 27c). Using primary culture neurons and HEK cells, we have shown that various proteins, including P62 and αS-pS-129 colocalize in the aggresome of αS inclusions and mediate toxicity in the presence of PD fibrils. In the presence of PD + SK-129 condition, we observed a significant decrease in the colocalization of P62 and αS-pS-129 in the aggresome of αS inclusions and rescue of the toxicity in HEK cells and primary culture neurons.

**Effect of SK-129 on αS aggregation mediated PD phenotypes in an in vivo model.** The antagonist activity of SK-129 against αS aggregation was tested in vivo using a *C elegans*-based PD model (NL5901). The ability of SK-129 to efficiently permeate cell membranes was confirmed by the parallel artificial membrane permeation assay (Fig. 5a) and confocal microscopy (using SK-129$_F$, Fig. 5b). The NL5901 strain is a well-established PD model that expresses WT αS-YFP in the body wall muscle cells[44,55] and PD phenotypic readouts include a gradual increase in inclusions (αS-YFP) in body wall muscle cells and a decline in motility during aging (Fig. 5c, e–g)[44,55]. The NL5901 strain was treated with 15 μM SK-129 at the larval stage and incubated with and without SK-129 for 9 days. The inclusions (αS-YFP) were counted manually using confocal microscopy. We observed a high number of inclusions (~33 inclusions/*C elegans*) (Fig. 5c, e and Movie S1); however, there was a substantial decline in inclusions in the presence of SK-129 (~8–9 inclusions/*C elegans*) (Fig. 5d, e and Movie S2), suggesting that SK-129 permeates the body wall muscle cell membrane and inhibits αS aggregation (Fig. 5d, e). The motility rate of the NL5901 strain decreases during the aging process as a consequence of αS inclusions. We utilized a newly developed WMicroTracker ARENA plate reader to measure the locomotion (overall activity counts) of NL5901 in the absence and presence of SK-129[56,57]. The overall activity of NL5901 displayed a gradual decline in the activity in comparison to the WT model of *C elegans* (N2) (Fig. 5f, g and Supplementary Fig. 28); however, NL5901 treated with 15 μM SK-129 at the larval stage resulted in a significant improvement in the overall activity (Fig. 5f, g and Supplementary Fig. 28). The overall activity of NL590 treated with SK-129 was closer to the N2 strain (Fig. 5f, g and Supplementary Fig. 28).

**The Antagonist Effect of SK-129 on the intracellular seed catalyzed aggregation of αS.** SK-129 was very potent antagonist of in vitro seed catalyzed aggregation of αS, both in cellular and primary culture neuronal models. However, in these models (cellular and neuronal), the solutions of αS aggregates (±SK-129) were prepared extracellular and then introduced to the cells or neurons to determine their ability to template the monomeric αS. Here, we aim to determine the antagonist activity of SK-129 against the seed catalyzed aggregation of αS in a novel intracellular assay using HEK cells. In this assay, the seeds of αS will be introduced to the HEK cells, followed by the introduction of

SK-129 to the cells. This assay will test the antagonist activity of SK-129 against the seed catalyzed aggregation of αS in an intra-cellular manner. To develop this assay, first, we assessed the total time required by αS seeds for the internalization into HEK cells. The HEK cells (expressing αS-$_{A53T}$-YFP) were exposed to αS seeds (0.125 μM) extracted from PD brain for various time points (0.5, 4, 8, 12, and 24 h), washed the cells, and incubated for a total of 24 h. The formation of αS inclusions was noticeable within 4 h of the treatment of cells with αS seeds. The number of inclusions were comparable (~20 inclusions/100 cells) in the case of 8, 12, and 24 h treatment of cells (Fig. 5h, i), which suggests that αS seeds were completely internalized in cells within 8 h. The antagonist activity of SK-129 against the intracellular seed cata-lyzed aggregation of αS was measured using the HEK cells treated with αS seeds for 8 h. A solution of SK-129 (10 μM) was added to HEK cells that were already treated with αS seeds (0.125 μM) for 8 h, followed by the incubation for an additional 16 h (total 24 h) (Fig. 5j). There was an abundance of inclusions in the absence of SK-129 after 24 h (~19 inclusions/100 cells, Fig. 5i); however, a low number of inclusions (~1–2 inclusions/100 cells) were observed in the presence of SK-129 (Fig. 5j). We also observed a gradual increase in inclusions in the presence of αS seeds for up to 4 days (Fig. 5j). However, in the presence of SK-129, a low number of inclusions was observed for up to 4 days (2–3 inclu-sions/100 cells, Fig. 5j). In addition, we also observed a gradual decrease in the cell viability of HEK cells treated with PD sample from day one to four (Fig. 4k). In marked contrast, the cell viability was significantly higher upto four days in the presence of SK-129 (Fig. 5k). The data clearly suggest that SK-129 is a potent antagonist of both intracellular de novo αS aggregation and the seed catalyzed aggregation of αS.

## Discussion

αS aggregation is one of the causal agents in PD pathologies, making it an enticing therapeutic target. However, the atomic-level understanding of the sequences that initiate αS aggregation is limited; therefore, strategies that identify aggregation-prone αS sequences could have significant therapeutic implications in the treatment of PD. We used OQs as a multipronged approach to investigate αS aggregation on a molecular level and to identify targets that are essential for the initiation of αS aggregation.

The study led to the identification of SK-129 as a potent inhibitor of de novo aggregation of αS under both in vitro and in vivo PD models. The data suggest that SK-129 stabilizes αS in an helical conformation by specifically interacting with distinct αS sequences towards the N-terminal of αS. We propose that SK-129 inhibits the aggregation of αS by either modulating the con-formation of monomeric αS into aggregation incompetent helical structure or SK-129 stabilizes the intermediate conformation of αS and inhibits the aggregation and rescue toxicity functions.

Deletion of the binding sites of SK-129 (αS sequences) from WT αS completely abolished the de novo and seed-catalyzed aggregation of αS. We postulate that the identified αS sequences are essential to initiate the aggregation and could be considered as novel therapeutic targets for the potent inhibition of αS aggre-gation. In groundbreaking findings, Eisenberg[13] and Radford[12] groups have identified αS sequences that initiate aggregation and they are in close proximity to the αS sequences identified from our study. More importantly, we have also validated these αS sequences by targeting them with foldamers, which led to the complete inhibition of αS aggregation and rescue of PD pheno-types in both in vivo and in vitro PD models. Our data suggest that the aggregation-prone αS sequences are potentially sampling helical conformation during αS aggregation; therefore, the design of helical mimetics complementing the chemical fingerprints of

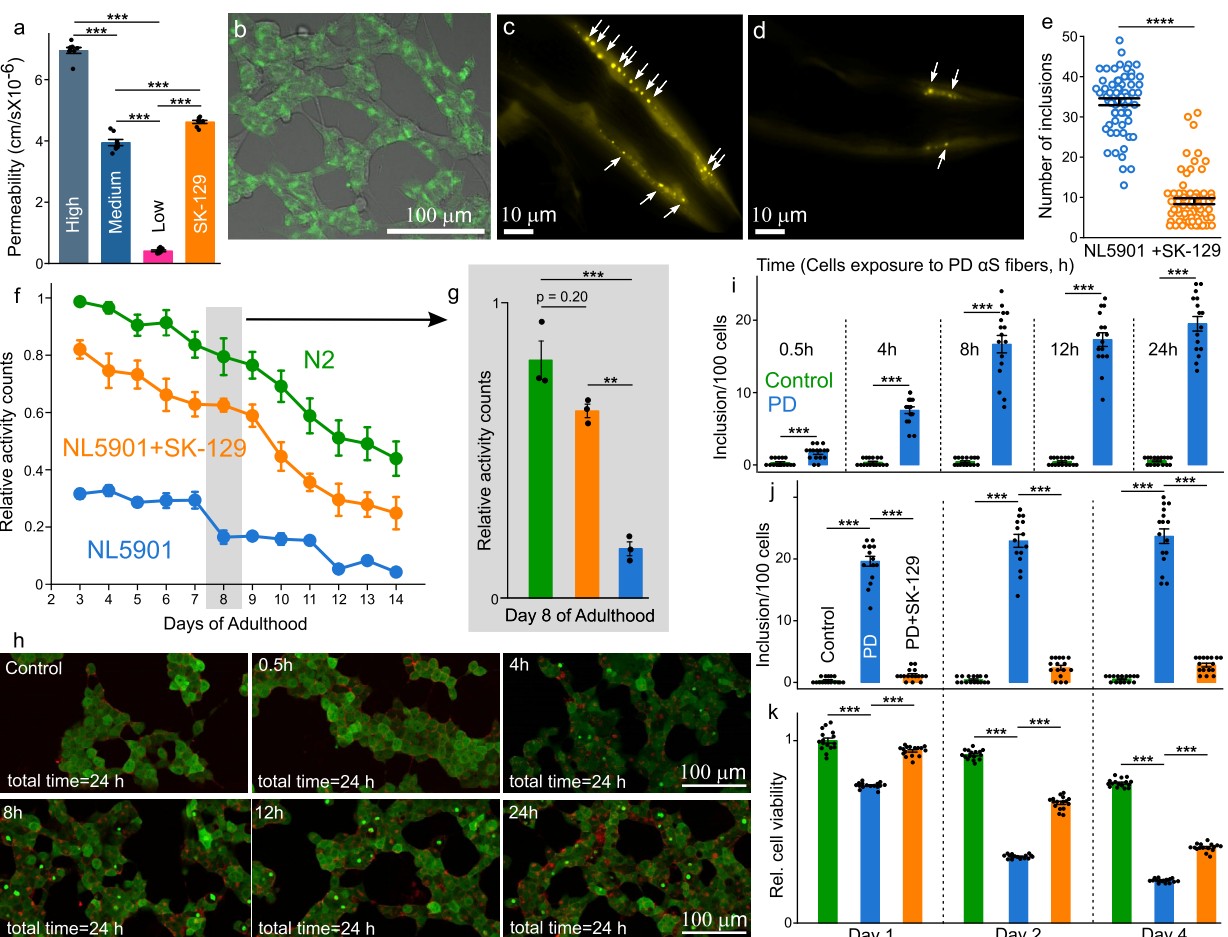

**Fig. 5 The intracellular inhibition of αS aggregation by SK-129 in PD models.** Assessment of cell permeability of SK-129 using PAMPA (**a**) and confocal microscopy (**b**). Confocal image of SH-SY5Y cells after treatment with 100 nM SK-129$_F$ for 15 h. The data were expressed as mean and the error bars report the s.e.m. (*n* = 4 independent experiments and each *n* consisted of two technical replicates). Confocal images (**c**, **d**) of αS-YFP inclusions (white arrows) in muscle cells of NL5901 (days of adulthood = 8 days) in the absence (**c**) and presence (**d**) of 15 μM SK-129. **e** The number of inclusions for experiment '**c**, **d**' for NL5901 worms in the absence and presence of SK-129 (days of adulthood = 8 days). The data were expressed as mean and the error bars report the s.e.m. (*n* = 4 independent experiments and each n consisted of at least 15 technical replicates). **f** The relative activity counts for 14 days of adulthood of N2 and NL5901 in the absence and presence of 15 μM SK-129. **g** A graphical representation with statistical analysis of the relative activity counts of N2 and NL5901 in the absence and presence of 15 μM SK-129 for day 8 of the adulthood. The data were expressed as mean and the error bars report the s.e.m. (*n* = 3 independent experiments and each *n* consisted of two technical replicates). **h** Confocal images and number of inclusions (**i**) of HEK cells incubated with PMCA sample from control and PD brain extract for indicated times. The HEK cells were washed after incubating them with the PMCA sample for various amounts of time. The number of inclusions (**j**) and relative viability (**k**) of HEK cells when they were incubated with the PMCA sample from PD brain extract for 8 h followed by washing, and further incubating HEK cells up to 4 days. A total of 100 HEK cells were examined to count the number of inclusions at four different locations in the eight-well plate for each experiment and it was repeated in 4 independent experiments. The data **i–k** were expressed as mean and the error bars report the s.e.m. (*n* = 4 independent experiments and each *n* consisted of four technical replicates). The statistical analysis was performed using ANOVA with Tukey's multiple comparison test. *$p < 0.05$, **$p < 0.01$, ***$p < 0.001$, ****$p < 0.0001$. Source data are provided as a Source Data file.

the helical conformation of αS sequences could lead to effective antagonism of αS aggregation and rescue of PD phenotypes.

We demonstrated that the binding sites of SK-129 (αS sequences) that initiate the de novo αS aggregation are also important for the seed-catalyzed aggregation of αS. The seed catalyzed aggregation requires the interaction of αS fibers with αS monomers to accelerate the aggregation. We have shown that SK-129 inhibits the seed catalyzed aggregation of αS. We surmise that the mode of action for SK-129 is a consequence of the interaction of SK-129 with the monomeric αS (towards N-terminal) and the conversion of the latter into a fiber-incompetent conformation, which consequently inhibits the seed catalyzed aggregation of αS. SK-129 was also a potent antagonist of de novo aggregation of αS (*C elegans* PD model) and seed catalyzed aggregation of αS (HEK cells) in the intracellular models. Based on our data, we propose

that SK-129 permeates the membrane, interacts with the intracellular monomeric αS, and modulates both the de novo aggregation of αS (*C elegans*) and the seed catalyzed aggregation of αS (HEK cells). A similar mode of action has been displayed by affibodies[58,59], which interact with monomeric αS and modulate it into a β-hairpin conformation. Similar to SK-129, the affibodies were potent inhibitor of both the de novo aggregation of αS and the seed catalyzed aggregation of αS[58,59].

Our atomic-level study suggests that SK-129 regulates the aggregation of αS under both de novo (lipid free) and lipid membrane conditions by binding to N-terminal sequences of αS, which are in close vicinity to the binding sites of molecular chaperones (Fig. 2, secB)[60]. Our NMR and CD data under de novo and lipid membrane conditions suggest that SK-129 regulates αS aggregation by shifting the equilibrium toward

non-aggregating and potentially functional αS, similar to molecular chaperones[60]. The chaperones have been shown to interact with the N-terminal region of αS and shift the conformational equilibrium towards the functional membrane-bound αS to maintain the cellular homeostatic balance [60]. SK-129 was able to inhibit the aggregation (de novo and lipid membrane conditions) and it was very effective in rescuing PD phenotypes in various cellular and in vivo PD models. If SK-129 was only able to inhibit αS aggregation and has interefer with the native function of αS, we would not have observed a significant rescue of toxic functions in various biological systems from PD phenotypes. Collectively, our data suggest that SK-129 potently inhibits αS aggregation without interfering with the native function of αS.

The modulation of αS aggregation by affibodies[58,59] and molecular chaperones[60] could be an attractive therapeutic intervention for PD; however, proteins/peptides are limited with poor cell permeability and poor enzymatic and conformational stability in biological milieus. Similarly, SK-129 has demonstrated chaperone/affibody-like ability to manipulate αS aggregation and it was able to efficiently rescue PD phenotypes in both in vitro and in vivo PD models. The intracellular antagonist activity of SK-129 in various PD models suggests that it possess good pharmaceutical properties, including good cell permeability, enzymatic stability, and structure stability because all of these properties are required for its activity against intracellular αS aggregation. The OQs have been previously shown to maintain potent antagonist activity against their therapeutic targets and demonstrated good cell permeability, structure stability, and enzymatic stability in the biological milieu. A PD mouse model-based study is underway to further assess the pharmaceutical properties and the antagonist activity of SK-129 against PD phenotypes. We are using a well-established mouse model of PD (αS_{A53T} transgenic line M83)[61,62]. This mouse model has been studied extensively because it mimics the PD pathologies[61,62]. Using this model, we will be able to assess the pharmacokinetics and pharmacodynamics properties of SK-129. Also, the mouse model study will be used to assess the ability of SK-129 to cross the blood–brain barrier. The mouse model study will be used to assess the antagonist activity of SK-129 against PD phenotypes. In addition, the PD mouse model will be treated with PD fibrils in the absence and presence of SK-129. The study will be used to assess the effect of SK-129 on the spreading and propagation of PD phenotypes facilitated by PD fibrils. We are optimistic that the pharmaceutical properties and the antagonist activity of SK-129 can be further optimized without sacrificing its overall conformation. The side chain functionalities of SK-129 scaffold can be conveniently modified synthetically without disturbing its overall conformation for further optimization of activity, which is often challenging with proteins/peptides. Additionally, the C- (COOMe of SK-129) and N-terminus (-NO₂ of SK-129) of SK-129 can also be modified to tune various pharmaceutical properties, including solubility and permeability etc. These manipulations in the chemical structure of SK-129 will not significantly alter its antagonist activity as we have seen with a fluorescent analog of SK-129 (SK-129_F), which has almost similar affinity to SK-129 against αS.

To the best of our knowledge, this is the first report that simultaneously led to the identification and validation of the chemical fingerprints of key sequences, which initiate αS aggregation and the targeting of these sequences completely abolishes αS aggregation.

## Methods

**Expression and purification of proteins**. The proteins, WT αS, Δ6-12 αS, and Δ47-53 αS were expressed and purified from the periplasm according to previously described protocol[63,64]. Briefly, the WT αS sequence cloned into pET11 vector (Addgene, Watertown, MA), Δ6-12 αS and Δ47-53 αS cloned in pET-21a(+)

(GenScript Biotech, Piscataway, NJ) were chemically transformed into *Escherichia coli* BL21(DE3) cells. Transformed cells were grown at 37 °C and shook at a rate of 200 rounds per minute (rpm) until the O.D. (optical density) was reached a value of 0.8. Protein expression was induced by adding isopropyl β-D-thiogalactoside (IPTG) at a final concentration of 1 mM. The induced cells were kept shaking at 200 rpm at 37 °C for 5 h. Cells were then collected by centrifugation (8217× g at 4 °C for 10 min) and resuspended in an osmotic shock buffer (30 mM Tris pH 7.2, 30% sucrose, 2 mM EDTA) and stirred for 15 min, similar to the reported protocol[64]. Subsequently, cells were collected again from the osmotic shock buffer by centrifugation (7177 × g for 10 min at 4 °C) and reconstituted in cold Milli-Q water and stirred for another 10 min. A solution of 5 mM MgCl₂ was added and stirred cells for an additional 5 min. Cells were removed by centrifugation at 5635×g for 10 min and the solution was boiled at 95 °C for 15 min for further purification[1]. The resulting protein precipitate was centrifuged (6000 × g for 20 min) and loaded on Bio-Scale Macro-Prep High Q ion-exchange column (Bio-Rad, Hercules, CA) (20 mM Tris pH 8.0, 25 mM NaCl, 1 mM EDTA). The protein was eluted with a high salt buffer (20 mM Tris pH 8.0, 1 M NaCl, 1 mM EDTA). The purified protein was buffer exchanged and concentrated in Milli-Q water using amicon ultra 3 K filters (MilliporeSigma, Burlington, MA). The concentration was determined using NanoDrop One ($\varepsilon_{280} = 5960\,M^{-1}\,cm^{-1}$) and lyophilized, then stored at −80 °C. Mutants Δ15-23 αS and Δ36-45 αS cloned into pET-21a(+) were transformed into T7 Express Iq competent cells. The cells were grown, induced, and collected as described above. The cell pellet was reconstituted in IEX A buffer (20 mM Tris pH 8.0, 25 mM NaCl, 1 mM EDTA) and boiled for 20 min at 95 °C. The insoluble fraction was removed by centrifugation at 6000 × g for 20 min and the soluble fraction was loaded on Bio-Scale Macro-Prep High Q ion-exchange column and purified as described above. The purified proteins (WT and mutants) were characterized using mass spectrometry (Mass spec facilities at the University of Colorado, Anschutz Medical Campus, CO and the University of Illinois at Urbana-Champaign mass spec facility, IL, Supplementary Fig. 22a–f) and the purity was confirmed with SDS-PAGE (Supplementary Fig. 21).

**ThT-based aggregation kinetic assay**. The aggregation kinetics of αS was monitored using ThT fluorescence assay ($\lambda_{ex} = 450$ nm and $\lambda_{em} = 485$ nm). To study the aggregation kinetics of αS, the ThT dye at a final concentration of 50 μM was added to 100 μM αS solution in the aggregation buffer (1 x PBS buffer) with and without various concentrations of SK-129 in a Costar black 96–well plate (Corning Inc., Kennebunk, ME). The small molecules were dissolved in dimethyl sulfoxide (DMSO; final DMSO concentration = 0.5%, v/v). The final volume in each well was 140 μL. The plate was incubated at 37 °C with 16 min shaking (434 rpm) and 44 min without shaking in an Infinite M200PRO plate reader (Tecan, Männedorf, Switzerland) and fluorescence was recorded after every hour. We used two methods to quantify the effect of molecules on the aggregation of αS; either we have reported the $t_{50}$ (time required to reach 50% fluorescence of ThT), which was extracted by fitting the ThT curve as a sigmoidal fit or the absolute ThT fluorescence intensity was reported. ThT-based aggregation kinetics were conducted three times and the reported $t_{50}$ (with or without ligands) is an average of three separate experiments. In the second method, we determined the final ThT fluorescence intensity for the aggregation of αS/Aβ₄₂(with or without ligands) as an average of three separate experiments. In this method, proteins (αS/Aβ₄₂) were aggregated in the absence and presence of various ligands at the indicated molar ratios similar to the above-mentioned conditions. The experiments were conducted three separate times and the reported ThT intensity was an average of three separate experiments. The ThT intensity was reported as relative intensity where the highest and lowest intensity were used from the protein sample and the control (ThT, DMSO, and buffer conditions only), respectively. The concentration of DMSO was kept constant (1%, v/v) in protein (αS/Aβ₄₂), control, and molecule solutions.

The aggregation kinetics of WT αS and αS variants were monitored and was carried out with a slightly modified protocol. A 100 μM monomeric WT αS/ αS variants solution (200 μL, in 1x PBS buffer) was placed into a ThermoMixer (Eppendorf, Hamberg, Germany) and shook for five days at a rate of 14,000 rpm and 37 °C. The aggregation of proteins was monitored at the indicated times (See the main manuscript). For each data point, a 5 μL protein solution (from 100 μM stock solution) was pipetted and diluted to a total of 100 μL (in 1x PBS buffer). To this solution, ThT dye (5 μL from a stock solution of 1 mM in 1x PBS buffer) was added and the solution was mixed well. The solution was transferred to a 96-well black plate and the ThT fluorescence intensity was measured using a 96-well plate reader. A solution of ThT dye in 1x PBS buffer was used as a control. For each protein (WT αS and αS variants), the ThT experiment was conducted in triplicate.

Seed-catalyzed αS aggregation assay was performed similarly to a previously described protocol[34]. Briefly, 100 μM monomeric WT αS solution (200 μL, in 1x PBS buffer) was placed into a ThermoMixer (Eppendorf, Hamberg, Germany) and shook for five days at a rate of 14,000 rpm and 37 °C. The solution was sonicated for 10 min and used as αS seeds. The seed-catalyzed aggregation of WT αS/ αS variants was performed by adding αS seeds (10% seeds, αS in monomeric concentration, v/v) into 140 μL of fresh 100 μM WT αS/ αS variants solution and aggregation was started with constant shaking at a rate of 14,000 rpm and 37 °C.

The aggregation of proteins was monitored at the indicated times (See the main manuscript) using the above-mentioned method.

The ThT aggregation assay for each protein (WT αS and αS variants) in the presence of SK-129 was carried out the above-mentioned method. SK-129 (from 10 mM stock solution in DMSO) was added at an equimolar ratio (1% DMSO, v/v) to the proteins at the start of the aggregation experiment. The concentration of DMSO was kept constant (1%, v/v) in protein (WT αS and αS variants), control, and protein+molecule solutions. The kinetic profiles of protein aggregation were processed using OriginPro software (Version 9.1). Kinetic curves were fit using a built-in sigmoidal fit in the OriginLab.

**Large unilamellar vesicles preparation**. The LUVs were prepared from a powder of 1,2-dioleoyl-sn-glycero-3-phospho-L-serine (sodium salt) (DOPS, Avanti Polar Lipids, Alabaster, AL). The powder was hydrated in 1x PBS buffer and vortexed for 15 min to make a 20 mM mixture of DOPS. The mixture of lipid in buffer was passed 21 times through a mini extruder (Avanti Polar Lipids, Alabaster, AL) using a polycarbonate membrane (Pore diameter = 100 nm, Avanti Polar Lipids, Alabaster, AL). The sizes of liposomes were confirmed using dynamic light scattering. The liposomes were used within a week of their preparation in various experiments.

**Circular dichroism spectroscopy**. The CD experiments were carried out on a JASCO J-1100 instrument. To study the kinetics of αS aggregation, a freshly prepared solution of 100 μM (or 70 μM) αS in 1x PBS buffer was used. The CD spectra were recorded from 260 nm to 195 nm at 0.5 nm intervals with 15 s averaging time and an average of three repeats. The αS solution was then aggregated at 37 °C with constant shaking (1400 rpm) for 7 days. The aggregated solution of αS was diluted to run the CD spectra. Similar CD experimental conditions were used to study the aggregation kinetics of various αS variants. To monitor the effect of SK-129 on αS aggregation, similar conditions were used in the presence of SK-129, except a solution of 1:1 molar ratio of SK-129: αS was used for CD experiments. The CD experiment in lipid membrane conditions was carried out with a solution of 30 μM αS in 375 μM LUVs (100 nm, DOPS) in the absence and presence of 30 μM SK-129. The CD spectrum of SK-129 was also recorded, which does not show any strong signal in the region of 190–260 nm and no interference was observed with SK-129 CD signal intensity. All CD experiments were conducted one time and the CD experiments between αS and SK-129 (or LUVs, DOPS) conducted at different stoichiometric ratios and a consistent trend of the change in the CD signals was observed as a function of stoichiometric ratios (of αS and SK-129/DOPS, LUVs), which supports the reproducibility of the data.

**αS and LUVs titration**. The binding affinity between αS and LUV's was determined according to previously described methods[12,38]. Briefly, 300 μL of a 40 μM αS solution in PBS buffer was placed in a 1 mm path length Quartz cuvette (Hellma, Plainview, NY) and the CD spectra was recorded at 30 °C, 1 nm data pitch and 1 nm band width using a Jasco-1100 CD spectrometer (Jasco, Easton, MD). Subsequently, increasing concentrations of LUV's (100 nm, DOPS) were titrated into the αS solution and mixed well before recording the CD scan. After each titration, 5 min was waited until the next measurement. The change in the CD signal (at wavelength = 222 nm) was plotted against the molar ratio LUVs to αS. The plot was fitted using one binding site model according to the earlier used methods[12,38].

**Transmission electron microscopy**. A solution of αS (600 μM) was incubated in 1x PBS buffer (150 mM NaCl, 2.7 mM KCl, 8 mM $Na_2HPO_4$, and 2 mM $KH_2PO_4$) in the absence and presence of SK-129 at an equimolar ratio at 37 °C and with constant shaking at 1200 rpm for 72 h. The aliquots (5 μL) of the solutions were applied on glow-discharged carbon-coated 300-mesh copper grids for 2 min and dried using tissue paper. The copper grids were negatively stained for 60 sec with uranyl acetate (0.75%, w/v). The micrographs were taken on an FEI Tecnai G2 Biotwin TEM at 80 kV accelerating voltages. The TEM experiments were repeated three times ($n = 3$) independently to ensure the reproducibility of the data.

**SDS-PAGE analysis**. The solution of αS (100 μM) in the absence and presence of SK-129 were prepared in the aggregation buffer (20 mM NaCl, 20 mM NaPi, pH 6.5) and kept at 37 °C and constant shaking at 1200 rpm until the aggregation of αS plateaued (four days). The solutions were separated into soluble and insoluble fractions of αS by centrifugation at 22,000 × g for 20 min. Afterwards, 2 x Laemmli protein sample loading buffer (Biorad, Hercules, CA) was added to the fractions and boiled at 95 °C for 5 min and ran on a 12% Mini-PROTEAN precast protein gel (Biorad, Hercules, CA). In addition, we also quantified SDS-PAGE gel band intensities using ImageJ software. The gel-based experiments were repeated three times ($n = 3$) independently to ensure the reproducibility of the data. All the uncropped gels (from Main manuscript and Supplementary Figures) have been included in the Supplementary Data File as Supplementary Fig. 29.

**αS seed extraction from the substantia nigra of the post mortem brains (control and PD)**. Fixed brain tissues were obtained from the Carroll A. Campbell,

Jr. Neuropathology Lab (CCNL) brain bank at the Medical University of South Carolina (Dr. Steve Carroll, Director) and from Dr. Greg Gerhardt's laboratory at the University of Kentucky. The seeds were extracted from the post mortem brain by following a published protocol with a slight modification[65]. The brain was sliced in 1 cm slabs as soon as possible to avoid long post mortem intervals and fixed free-floating in a solution of 4% paraformaldehyde in 1x PBS buffer (4% paraformaldehyde in 1x PBS buffer). All brain tissues used in the study had a post mortem interval (PMI) of less than 12 h. Following fixation, tissue was transferred to a cryoprotectant solution (30% glycerol, 30% ethylene glycol, and 40% 1x PBS buffer, v,v) and kept at −20 °C until dissection. A total of Sixteen brain regions were dissected after fixation and processed for neuropathological diagnosis of PD according to a published protocol[65]. A diagnosis was conducted, which included staining with H & E, Bielshowsky's silver stain, as well as p-Tau, Amyloid-β, and αS immunohistochemistry (See main manuscript Fig. 4b, c). From the total tissue amount, ~100 mg of fixed *substantia nigra* tissue was homogenized using a Bio-Gen PRO200 Tissue Homogenizer (PRO Scientific Inc., Oxford, CT) in 1x PBS buffer (10%, w/v). The homogenate was centrifuged at a rate of 19,000 × g for 12 min. The supernatant was discarded, and the pellet was homogenized again in 1x PBS buffer supplemented with Triton-X 100 (1%, v/v). The solution was centrifuged (19,000 × g for 12 min) to remove soluble components and the pellet was reconstituted in 1x PBS buffer with 1% Triton X-100 (v,v) and stored at −80 °C prior to use in various experiments.

**Protein misfolding amplification assay**. The PMCA was performed according to the previously described method[66]. A lyophilized αS powder was dissolved in 1x PBS buffer to a final concentration of 90 μM. Subsequently, 60 μL of the 90 μM αS solution was placed in 200 μL Polymerase Chain reaction (PCR) tubes and the mixture was subjected to 24 h cycles of 1 min shaking (1200 rpm) and 29 min incubation at 37 °C. Every 24 h, 1 μL of PMCA incubated sample was transferred to a fresh soluble monomeric αS solution, which was repeated for five days. For the preparation of PMCA samples of αS in the presence of SK-129, the samples were prepared by adding SK-129 to maintain a molar ratio of 1:1 (αS:SK-129). Control samples were prepared with an equal volume of DMSO (0.9%) as used in the case of SK-129. All experiments were performed in triplicates.

For the PMCA assay for αS seeds extracted from PD brain, we have used similar conditions to αS seeds from recombinant αS with slight modification. The αS seeds extracted from PD brain were added at 5% (v/v) to freshly prepared monomeric αS solution (60 μL and 90 μM) and the assay was carried out as described above for the recombinant αS seeds where we used a 1 μL solution for each cycle for up to five cycles.

**Proteinase K digestion of PMCA samples**. A 50 μg/ml solution of PK (IBI Scientific, Dubuque, IA) in the digestion buffer (10 mM Tris pH 8.0, 2 mM $CaCl_2$) was diluted 10 times in 30 μL of PMCA solutions and incubated for 30 min at 37 °C. Subsequently, the sample was diluted 2 times in SDS Protein Gel Loading Dye 2 × (Quality Biological, Gaithersburg, MD) and loaded on Mini-PROTEAN TGX Stain-Free Protein Gel (BioRad, Hercules, CA). The gel was stained with Fairbanks staining method and then imaged using ChemiDoc MP (BioRad, Hercules, CA).

**Primary rat hippocampal neuron culture**. Pregnant Sprague Dawley Rats (72-85 days old, mixed male and female) were purchased from Charles River Laboratories (Strain Code 400) and maintained at the University of Denver Animal facility (AAALAC accredited). All animal protocols and experiments were approved by the University of Denver Animal Care and Use Committee.

The rat embryos (both male and female) at embryonic day 18 were used to prepare the primary neurons according to previously published protocol[67]. After removing all the meninges, the hippocampi were isolated from the fetal rat brain and kept in the dissection solution (1x HBSS, 10 mM HEPES buffer, 5 μg/mL Gentamicin, pH 7.3, Thermo Fisher Scientific, Waltham, MA). The hippocampi were minced and treated in the dissection solution containing 20 U/ml Papain (Worthington Biochemical Corp., Lakewood, NJ) and triturated in 50 μg/ml Dnase I (Sigma-Aldrich, St.Louis, MO). The isolated cells were plated on 1 mg/mL Poly-L-Lysine (Sigma-Aldrich, St.Louis, MO) coated μ-slide eight-well plate (Ibidi, Munich, Germany) at 200,000 cells/mL in neuron plating medium, which includes Minimum Essential Media (MEM)(Thermo Fisher Scientific, Waltham, MA) supplemented with 5% FBS (Thermo Fisher Scientific, Waltham, MA) and glucose (Sigma-Aldrich, St.Louis, MO). After neurons adhered, the neuron plating medium was replaced with Neurobasal media (Thermo Fisher Scientific, Waltham, MA) with 0.3 C GlutaMAX (Thermo Fisher Scientific, Waltham, MA) and 1 X B-27 (Thermo Fisher Scientific, Waltham, MA) and neurons were maintained at 37 °C and 5% $CO_2$(g).

The fiber solution made from brain seeds via PMCA assay (5the cycle) in the absence and presence of SK-129 were added to the primary culture neurons in the eight-well plate. The seeds were added at a concentration of 1 μM (αS in monomer concentration) on DIV 10 (days in vitro). The primary cultured neurons were incubated with various conditions for DIV 21 before carrying out experiments, including lactate dehydrogenase (LDH) release assay and immunocytochemistry.

**LDH release assay to assess the neurotoxicity of primary neurons**. The LDH assay was performed on the primary neurons treated with various conditions and incubated for 21 DIV. The release of LDH was measured using a Cytotox 96 Non-Radioactive Cytotoxicity Assay kit (Promega, Madison, WI) by following the instructions provided by the manufacturers. After the neurons were incubated with various conditions for DIV 21, the media was collected from the wells (300 μL). The media for each condition (50 μL/each well) was transferred to a flat-bottom 96-well cell culture plate (Costar, Kennebunk, ME). To this media solution, 50 μL of the Cytotox reagent was added to make it a total of 100 μL solution and mixed gently with a pipette. The absorbance of the media in the wells was measured at 490 nm using a 96-well plate reader.

**Immunocytochemistry of primary rat hippocampal neurons**. The primary neurons treated under various conditions in a μ-slide eight-well plate were washed three times with an ice-cold 1x PBS buffer and fixed with 4% paraformaldehyde for 10 min at room temperature (RT). Afterward, neurons were permeabilized with PBST buffer (0.15%, v/v, Triton-X 100 in 1x PBS buffer) for 10 min and blocked with 5% Bovine serum albumin (BSA) in PBST buffer. Subsequently, the fixed primary neurons were stained with various primary antibodies including, LB biomarkers, αS-pS-129 (stains phosphorylated residue 129 of αS), p62 (stains autophagosome vesicles), and TOM20 (stains mitochondria) and ThS, a dye that specifically stains αS aggregates. All primary antibodies were diluted to a ratio of 1:1000 (v,v) in a 5% BSA solution in the PBST buffer. The primary neurons were incubated with the primary antibodies overnight at 4 °C and then washed five times with the PBST buffer. The primary neurons were then incubated with secondary antibodies (dilution to a ratio of 1:1000 (v,v) in a 5% BSA solution in the PBST buffer) for 1 h at RT. Afterward, the eight-well plate was washed with 1x PBS buffer (three times) to remove any excess of the antibodies. Subsequently, the cells were stained with 300 μL/well of Thioflavin S in the PBST buffer (1%, w/v) (Sigma-Aldrich, St. Louis, MO) for 10 mins and washed five times with PBST buffer and then used for confocal imaging. The primary and secondary antibody specifications are included in the Reporting Summary (Antibodies section).

**Confocal imaging of the primary culture neurons**. The eight-well plate with neurons treated under different conditions was fixed and stained with various biomarkers were then imaged using confocal microscopy. The confocal imaging was performed on an Olympus Fluoview FV3000 confocal/2-photon microscope, using a 20×Plan-Apo/1.3 NA objective with DIC capability. The confocal images of the primary neurons were then processed using the OlympusViewer in ImageJ processing software. The confocal imaging experiments were repeated four times independently to ensure the reproducibility of the data.

**Fluorescence polarization titrations**. All Fluorescence Polarization (FP) experiments were conducted on a Varian Cary Eclipse fluorometer (Agilent, Santa Clara, CA) equipped with a polarizer. The stock solutions of αS, 1 mM or 100 μM in 1x PBS buffer, were serially added into a solution of 10 μM SK-129$_F$ (300 μL in 1x PBS buffer) while stirring in a quartz cuvette (Fireflysci, Staten Island, NY). All FP measurements were conducted in triplicates in 20 mM Tris, 100 mM NaCl, pH 7.2 at 20 °C. The excitation and emissions wavelengths were set at 490 nm and 520 nm, respectively. Each titration was equilibrated for five min before measurements were taken for the next addition of αS solution. The addition of αS solution was continuous until no more change in the FP signal was observed. Similar conditions were used for the titration of various αS mutants and Aβ$_{42}$ peptide against SK-129$_F$. A stock solution of Aβ$_{42}$ (1 mM or 100 μM in 1x PBS buffer on ice) was sequentially added to a solution of 10 μM SK-129$_F$ (300 μL in 1x PBS buffer) until no more change in the FP signal was observed. The polarization of each point was calculated using Eq. 1:

$$FP = \frac{I_{\parallel} - I_{\perp}}{I_{\parallel} + I_{\perp}} \qquad (1)$$

Where $I_{\parallel}$ is fluorescence intensity parallel to the excitation plane and $I_{\perp}$ is fluorescence intensity perpendicular to the excitation plane. The plot between the relative change in the FP signal against the concentration of the protein (αS/ Aβ$_{42}$) was fit using one site binding model and $K_d$ between SK-129$_F$ and αS (αS mutants/Aβ$_{42}$) was extracted from the fit. For displacement titration, a 10 μM SK-129$_F$ solution was saturated with 10 mole equivalents of αS (300 μL in 1x PBS buffer). To this solution, a stock solution of SK-129 (1 mM or 10 mM) was serially added to displace SK-129$_F$ from the αS-SK-129$_F$ complex. The solution of SK-129 was continuously added until no more change in the FP signal was observed. A plot between the relative change in the FP signal against the concentration of SK-129 was fit using a one-site competitive binding model and K$_d$ between SK-129 and αS was extracted from the fit.

**Parallel artificial membrane permeability assay**. A parallel artificial membrane permeability assay (PAMPA) Kit (BioAssay Systems, Hayward, CA) was used to measure the membrane permeability of SK-129 according to the manufacturer's protocol. Briefly, a 4% lecithin solution (LS) was prepared in dodecane and solubilized with constant sonication for 20 min. Then, 5 μL of LS was placed on the donor plate membranes. A 300 μL 1x PBS buffer was applied to the acceptor plate.

The solutions of SK-129 and various permeability controls (Highly soluble, medium soluble and low soluble) molecules were added to donor plates (200 μL and 500 μM). The donor plate was placed in the acceptor plate and incubated at RT for 18 h. Then the solutions were removed from the acceptor plate and placed in a clear-bottom 96-well plate (Corning Inc., Corning, NY) and absorbance was recorded at 360 nm for SK-129 and 275 nm for standards. The permeability was calculated using Eq. 2:

$$P_e = C \times \left[ -\ln\left(1 - \frac{OD_A}{OD_E}\right) \right] cm/s \qquad (2)$$

Where the permeability rate $C$ is $7.72 \times 10^{-6}$, $OD_A$ is the absorbance of acceptor solution and $OD_E$ is absorbance of equilibrium standard[14].

**The 3-(4,5-dimethylthiazol-2-yl)-2,5-diphenyltetrazolium bromide (MTT)-Based Cytotoxicity Reduction Assay (SH-SY5Y cells)**. An MTT assay was conducted to assess the effect of SK-129 on the cytotoxicity mediated by αS aggregation in SH-SY5Y cells. The cells were cultured in phenol red-free Dulbeco's Modified Eagle Medium (DMEM) with 10% Fetal Bovine Serum (FBS) and 1% penicillin-streptomycin (Pen/strep) at 37 °C and 5% CO$_2$(g). The cells were plated in a clear, flat bottom 96-well cell culture plate (Costar, Kennebunk, ME) using a density of 10,000 cells per well with more than 90% cell viability. After incubating for 24 h, the media was aspirated, and 100 μL of fresh OptiMEM (Fisher Scientific, Pittsburgh, PA) was added, which contains the aggregated solution of 25 μM of αS in the absence and presence of various ligands at various stoichiometric ratios. The samples were then incubated for an additional 24 h at 37 °C and 5% CO$_2$(g), followed by the addition of MTT dye (10 μL per well, prepared in 1x PBS buffer, 5 mg/mL). The plates were covered in aluminum foil and incubated again for 3 h. The solution in each well was carefully removed without disturbing the formazan crystals and replaced with 100 μL of DMSO to dissolve the formazan crystals. Subsequently, the plate was shaken in a 96-well plate reader for 5 min before measuring the absorbance at 570 nm. The cell viability was reported on a scale of 100%, using the control wells with regular media as 100% viability and the wells with 10% DMSO (v/v) as 0% viability.

**General method for the transfection of HEK cells with αS fibrils using Lipofectamine solution**. The Lipofectamine solution (Lipofectamine+P3000 reagent, Thermo Fisher Scientific, Waltham, MA) was diluted to a ratio of 1:20 (v/v) in the OptiMEM (Fisher Scientific, Pittsburgh, PA) media. Simultaneously, the αS fiber solution (Stock solution conc. = 100 μM) was diluted in OptiMEM media to the desired conc. used for each assay. The αS fiber solution was sonicated for 10 min at r.t., followed by the addition of the Lipofectamine solution (in the OptiMEM media) at 1:1 ratio. Subsequently, this solution was incubated for another 10 min and then added to the HEK cells media with a dilution factor of 10 (10 μL of the combined αS fiber solution+Lipofectamine solution, in 90 μL HEK cells media).

**The MTT assay (HEK293 cells)**. The HEK293 cells that stably express αS-$_{A53T}$-YFP and αS-YFP were grown in DMEM with 10% FBS and 1% pen/strep and cultured in an incubator at 37 °C and 5% CO$_2$(g). A total of 60,000 cells per well in 300 μL media were plated in a μ-slide eight-well plate (Ibidi, Munich, Germany) and incubated for 24 h to adhere to the plate. After 24 h, the media was aspirated and 300 μL of OptiMEM (Fisher Scientific, Pittsburgh, PA) containing αS fibrils, 0.125 μM and 7.5 μM of the brain sample and recombinant protein, respectively, were added in the absence and presence of SK-129 at an equimolar ratio in the presence of Lipofectamine 3000 (Thermo Fisher Scientific, Waltham, MA). Next, the plate was incubated for 24 h, followed by the addition of 30 μL of 1x PBS buffer containing MTT dye (in 1x PBS buffer, 5 mg/mL) to each well. The plates were wrapped in aluminum foil and incubated for 3 h. After 3 h, all liquid was aspirated carefully without disturbing the formazan crystals. To each well, 300 μL of DMSO was added to dissolve the crystals. Once dissolved, the DMSO solution was transferred to a clear 96-well plate (100 μL/well). The absorbance was read on a 96-well plate reader at 570 nm. The cytotoxicity of the HEK cells was monitored for up to 4 days.

The seeds were generated by aggregating recombinant αS (100 μM) in the absence and presence of SK-129 at an equimolar ratio under the ThT aggregation kinetic conditions for seven days. In the case of seeds from the brain, we used a PMCA sample (5th cycle) generated from the brain seeds in the absence and presence of SK-129 at an equimolar ratio.

**Confocal imaging of HEK cells**. The HEK cells expressing αS-$_{A53T}$-YFP or αS-YFP (200,000 cells/mL) were plated in a μ-slide eight-well plate (Ibidi, Gräfelfing, Germany) (300 μL/well) and incubated at 37 °C and 5% CO$_2$ (g), and allowed to adhere to the plate for 24 h in complete media (DMEM, 10% FBS, 1% pen/strep). After 24 h, the media was aspirated and 300 μL of OptiMEM containing αS fibrils, 0.125 μM and 7.5 μM of the PD brain sample (PMCA, 5th cycle) and recombinant protein in the presence of Lipofectamine 3000, respectively were added. The plate was incubated for 48 h after the addition of fibers. The HEK cells were treated for 1 h with a mixture of Hoechst 33342 dye solution (3 μL/well from 1 mg/mL solution in 1x PBS buffer) and wheat germ agglutinin alexa fluor 633 conjugate (3 μL/well from 1 mg/mL solution in 1x PBS buffer) to stain nuclei and the plasma

membrane, respectively of the cells. The HEK cells were washed with the 1x PBS buffer (four times) to remove excess traces of dyes and used for the live-cell confocal imaging. The confocal imaging was performed on an Olympus Fluoview FV3000 confocal/2-photon microscope, using a 20×Plan-Apo/1.3 NA objective with DIC capability. The confocal images of the HEK cells were processed using the OlympusViewer in ImageJ processing software.

Similar conditions were used to monitor the effect of SK-129 on the aggregation of αS in HEK cells. A solution of 100 μM αS was aggregated for seven days in 1x PBS buffer in the absence and presence of SK-129 at an equimolar ratio. The fibers of αS in the absence and presence of SK-129 in the presence of Lipofectamine 3000 were used in HEK cells. For PD brain samples, the PD brain seeds were used in the PMCA assay with fresh αS sample in the absence and presence of SK-129 at an equimolar ratio. The PMCA samples from the fifth cycle in the absence and presence of SK-129 were used for the HEK cells. The confocal imaging experiments were repeated four times independently to ensure the reproducibility of the data.

**Immunofluorescence staining and confocal imaging of HEK cells**. The HEK cells (αS$_{A53T}$-YFP) were plated (300 μL/well) in a μ-slide eight-well plate (Ibidi, Gräfelfing, Germany) and incubated (24 h at 37 °C and 5% CO$_2$). The cells were then transfected with αS fibrils (7.5 μM in monomeric conc. of αS) aggregated in the absence and presence of SK-129, using lipofectamine and incubated for 24 h as described in the previous protocol. The transfection was confirmed by the appearance of puncta within the plated cells using an Axio Observer microscope. The cells were then fixed with 4% paraformaldehyde for 10 min and subsequently washed with PBS (3×). The paraformaldehyde was then replaced with PBS containing 0.15% Triton X-100 for 10 min and washed with PBS (3×). A PBS solution containing 1% (w/v) BSA (Thermo-Fischer Scientific, Rockford, IL) and 0.1% (v/v) Tween-20 (Sigma-Aldrich, St. Louis, MO) was added and incubated at r.t. for an additional 30 min then washed with PBS (3×). Next, the cells were stained with anti-α-Synuclein Phospho Ser129 (phosphorylated αS at serine residue 129) mouse antibody (BioLegend, San Diego, CA) or anti-p62 (Millipore Sigma, Burlington, MA) for 1 h and washed with PBS (3×). The cells were then treated with Donkey anti-rabbit tagged with Alexa Fluor Plus 647 (for anti-p62) secondary antibody (Invitrogen, Rockford, IL) or goat anti-mouse tagged with Alexa Fluor 680 (for anti-pS129) secondary antibody for 1 h and washed with PBS (3×). Lastly, the cell nuclei were stained with DAPI (Cayman Chemical Company, Ann Arbor, MI) for 10 min and washed with PBS (2×). All primary and secondary antibodies were diluted to a ratio of 1:1000 (v/v) in TBST buffer containing 5% BSA. Confocal imaging was conducted on an Olympus Fluoview (FV3000 confocal/2-photon microscope). The images produced by the confocal microscope were analyzed on OlympusViewer in ImageJ processing software.

**Inclusion quantification assay in HEK cells**. The αS inclusions in HEK cells (expressing endogenous αS$_{-A53T}$-YFP and αS-YFP) were quantified using three methods.

**Method 1 (confocal imaging)**. We used confocal microscopy imaging to manually count the endogenous inclusions (of αS-$_{A53T}$-YFP and αS-YFP). The HEK cells (200,000 cells/mL) were plated in an μ-slide eight-well plate (300 μL/well) and incubated at 37 °C and 5% CO$_2$ (g), and allowed to adhere to the plate for 24 h in media (DMEM, 10% FBS, 1% pen/strep). After 24 h, the media was aspirated and 300 μL of OptiMEM containing αS fibrils, 0.125 μM and 7.5 μM of the brain sample and recombinant protein (in the absence and presence of SK-129 at an equimolar ratio) in the presence of Lipofectamine 3000, respectively were added. The plate was incubated for 48 h after the addition of fibers. The cells were treated with a Hoechst 33342 dye solution for 1 h by adding 3 μL to each well (from 1 mg/ml solution in 1x PBS buffer). The Hoechst 33342 solution was carefully mixed with media by pipetting up and down a few times. The cells were then carefully washed with 1x PBS buffer three times for confocal imaging. The confocal imaging was performed on an Olympus Fluoview FV3000 confocal/2-photon microscope, using a 20×Plan-Apo/1.3 NA objective with DIC capability. The confocal images of the HEK cells were processed using the OlympusViewer in ImageJ processing software. The counting of inclusions was carried out for six different experiments and for each experiment, 100 cells were counted from at least four different locations from the eight-well plate. The cells were counted manually by counting the number of Hoechst 33342 dye-stained nuclei in the cells. Also, the inclusions in the cells were counted manually at different locations in the eight-well plate.

**Method 2 (ProteoStat dye staining)**. The inclusions in HEK cells (αS$_{-A53T}$-YFP and αS-YFP) were quantified using the ProteoStat Protein Aggregation Assay Kit (Enzo Life Sciences, Farmingdale, NY). The HEK cells (200,000 cells/mL) were plated in an μ-slide eight-well plate (300 μL/well) and incubated at 37 °C and 5% CO$_2$ (g), and allowed to adhere to the plate for 24 h in media (DMEM, 10% FBS, 1% pen/strep). After 24 h, the media was aspirated and 300 μL of OptiMEM containing 0.125 μM concentration of αS fibrils, which are generated alone and in the presence of SK-129, were added to the plate in the presence of Lipofectamine 3000. The plate was incubated for 48 h after the addition of fibers. After incubating, the media was aspirated (saved) and 150 μL of detachin was added to each well and incubated for 5–10 min until cells were completely detached from the flask surface.

A 250 μL solution of 1x PBS was then added to each well. The total of 400 μL solution was collected in eppendorf tubes for each condition and centrifuged for 5 min at a rate of 2000 × g. The supernatant was removed, and the cell pellets were washed with the 1x PBS buffer (100 μL, two times). To the cell pellets, 500 μL of 4% paraformaldehyde was added to each condition and the tubes were incubated in an ice bath for 30 min. The cell solution was centrifuged and washed with the 1x PBS buffer (500 μL, two times), then 500 μL of 0.1% Triton (1x PBS buffer, v/v) was added to the cells, and they were incubated for 20 min on ice. The cell solution was centrifuged and washed with 1x PBS (500 μL, two times). Subsequently, 375 μL of the ProteoStat dye was added to each condition (1350 μL 1x PBS buffer, 150 μL 10 × assay buffer, 1.5 μL ProteoStat dye) and wrapped in aluminum foil and incubated at RT for 20 min. The cell solution was centrifuged, and the supernatant was aspirated. The cells were then homogenized with 400 μL of 1x PBS buffer and transferred to a black 96–well plate (Corning Inc., Corning, NY) (100 μL/each well and four wells per condition) and the fluorescence was measured using a 96-well plate reader (λ$_{ex}$ = 550 nm, λ$_{em}$ = 600 nm).

**Method 3: FACS (fluorescence-activated cell sorting) analysis**. For flow cytometry experiments, the HEK cells expressing αS$_{-A53T}$-YFP or αS-YFP proteins (200,000 cells/mL) were treated with different conditions (αS fibrils with and without SK-129 in the presence of Lipofectamine 3000), fixed, and stained with Proteostat dye as described in the previous method. The HEK cells were analyzed by flow cytometry using the Sony cell sorter (SH800, San Jose, CA) using a 488 nm laser and 525/50 FL2 (YFP) and 600/60 FL3 (Proteostat dye) filters. The gating was created based on the fluorescence intensity of the control cells (No αS fibrils) without seeding. For each sample, 10,000 cells were counted, analyzed, and plotted using Cell Sorter Software (Version 1.7, LE-SH800 Series, Sony, San Jose, CA). To further quantify the inclusions in HEK cells treated with different conditions, the histograms were divided into two parts on the x-axis. The x-axis represents the Proteostat signal intensity in HEK cells, which was detected in a single channel with 600/60 FL3 (Proteostat dye) filters. The flow cytometry experiments were conducted one time each for HEK cells expressing αS$_{-A53T}$-YFP or αS-YFP proteins. A similar trend was observed on the intracellular aggregation of both proteins (αS$_{-A53T}$-YFP or αS-YFP) facilitated by exogenously added αS fibers in the absence and presence of SK-129. These results support the reproducibility of the data form the flow cytometry experiments.

**Time-dependent effect of PD brain αS seeds on the monomeric αS in HEK cells**. The HEK cells (200,000 cells/mL) were plated in an μ-slide eight-well plate (300 μL/well) and incubated at 37 °C and 5% CO$_2$ (g), and allowed to adhere to the plate for 24 h in media (DMEM, 10% FBS, 1% pen/strep). After 24 h, the media was aspirated and 300 μL of OptiMEM containing αS fibrils, 0.125 μM of the PMCA sample (Cycle 5$^{th}$) was added to the eight-well plate in the presence of Lipofectamine 3000. The HEK cells were incubated with αS fibrils for various time points, including 0 h (control), 2 h, 4 h, 8 h, 12 h, and 24 h (2 wells/condition, 300 μL/well) and then the αS fibrils were washed with HEK cells using OptiMEM (three times). The eight-well plate was incubated for a total of 24 h after the addition of αS fibrils. The HEK cells were treated for 1 h with a mixture of Hoechst 33342 dye solution (3 μL/well from 1 mg/mL solution in 1x PBS buffer) and wheat germ agglutinin alexa fluor 633 conjugate (3 μL/well from 1 mg/mL solution in 1x PBS buffer) to stain nuclei and plasma membrane, respectively of the cells. The dye solutions were mixed carefully with the media by pipetting up and down for a few times. The HEK cells were washed with the 1x PBS buffer (four times) to remove excess traces of dyes and used for the live-cell confocal imaging. The confocal imaging was performed on an Olympus Fluoview FV3000 confocal/2-photon microscope, using a 20×Plan-Apo/1.3 NA objective with DIC capability. The confocal images of the HEK cells were processed using the OlympusViewer in ImageJ processing software. The counting of inclusions was carried out for six different experiments and for each experiment, 100 cells were counted from at least four different locations from the eight-well plate. The cells were counted manually by counting the number of Hoechst 33342 dye-stained nuclei in the cells. The inclusions in the cells were counted manually at different locations in the eight-well plate.

**Effect of SK-129 on PD brain αS seeds mediated intracellular prion-like spread of αS (inclusions and cytotoxicity)**. The HEK cells (200,000 cells/mL) were plated in an μ-slide eight-well plate (300 μL/well) and incubated at 37 °C and 5% CO$_2$ (g), and allowed to adhere to the plate for 24 h in media (DMEM, 10% FBS, 1% pen/strep). After 24 h, the media was aspirated and 300 μL of OptiMEM containing αS fibrils, 0.125 μM of the PMCA sample (Cycle 5th) was added to the eight-well plate in the presence of Lipofectamine 3000. The HEK cells were incubated with αS fibrils for 8 h followed by the addition of 10 μM SK-129 for 300 μL media in each well (0.3 μL of 10 mM stock solution in DMSO, total DMSO = 0.1% v,v). The eight-well plate was incubated for a total of 24 h. The eight-well plate after 24 h was used for both confocal imaging and for the cytotoxicity measurements using MTT assay (2 wells/condition, 300 μL/well). In each experiment, 2 wells were used for each condition, including control (only media and DMSO), fibrils, and fibrils+SK-129.

**Confocal imaging**. The HEK cells in the eight-well plate were treated with a Hoechst 33342 dye solution for 1 h by adding 3 μL to each well (from 1 mg/ml solution in 1x PBS buffer). The Hoechst 33342 solution was carefully mixed with media by pipetting up and down for a few times. The cells were then carefully washed with 1x PBS buffer three times for confocal imaging. The confocal imaging was performed on an Olympus Fluoview FV3000 confocal/2-photon microscope, using a 20×Plan-Apo/1.3 NA objective with DIC capability. The confocal images of the HEK cells were processed using the OlympusViewer in ImageJ processing software. The counting of inclusions was carried out for six different experiments and for each experiment, 100 cells were counted from at least four different locations from the eight-well plate. The cells were counted manually by counting the number of Hoechst 33342 dye-stained nuclei in the cells. The inclusions in the cells were counted manually at different locations in the eight-well plate. A similar experiment was also conducted for HEK cells after incubating them for longer times (2 and 4 days).

**Cytotoxicity assay (MTT-reduction based assay)**. The HEK cells in the eight-well plate were treated with 30 μL of 1x PBS buffer containing MTT dye (5 mg/mL in 1x PBS buffer) to each well. The plates were wrapped in aluminum foil and incubated for 3 h. After 3 h, all liquid was aspirated carefully without disturbing the formazan crystals. To each well, 300 μL of DMSO was added to dissolve the crystals. Once dissolved, the DMSO solution was transferred to a clear 96-well plate (100 μL/well). The absorbance was read on a 96-well plate reader at 570 nm. The cytotoxicity assay was conducted up to four days from when the protein conditions were added to the cells. A similar experiment was also conducted for HEK cells after incubating them for longer times (2 and 4 days).

**2D HSQC NMR spectroscopy**. Two-dimensional $^1$H-$^{15}$N HSQC NMR experiments were performed on a 600 MHz Bruker instrument equipped with a triple resonance HCN cryoprobe. Uniformly labeled $^{15}$N-αS (>95% purity from SDS-PAGE) was purchased from rpeptide (Bogart, GA). One mg of αS powder was dissolved in 970 μL of milli-Q water (0.2 μM filter) to make a final concentration of 70 μM in 1x PBS buffer, which was eventually taken into another buffer of 20 mM NaPi, pH 6.4. The αS solutions were divided into 350 μL aliquots, lyophilized, and stored at −80 °C until further use. The concentration of each aliquot was determined spectroscopically at 280 nm using an extinction coefficient of 5960 $M^{-1}cm^{-1}$. The 2D HSQC NMR experiments were carried out in 20 mM NaPi, pH 6.4, and by maintaining a solution ratio of 90:10 (H2O:D2O) at 15 °C. For each 2D HSQC NMR experiment, a freshly prepared solution of $^{15}$N-αS was used and the experiments were carried out at 15 °C to avoid potential complications from the amyloid formation. For the HSQC experiment in the presence of SK-129, a stock solution of 10 mM SK-129 was prepared in DMSO (pure, HPLC grade). Under the NMR conditions used here, αS was exclusively found in the monomeric state. For HSQC NMR in the presence of SK-129, a fresh solution of $^{15}$N-αS was prepared at a concentration of 70 μM (350 μL) in the presence of 70 μM and 140 μM concentration of SK-129 (stock solution of 10 mM in DMSO). The solution was mixed well before starting the HSQC NMR experiments. The $^{15}$N-αS sample was diluted to 0.8% in the presence of SK-129 at an equimolar ratio. For $^1$H-$^{15}$N HSQC NMR experiments, data for the $^1$H and $^{15}$N frequencies were acquired using 1024 and 512 points, respectively. Apodization was achieved in the $^1$H and $^{15}$N dimensions using a sine square function shifted by 90°. The 2D HSQC NMR spectra were processed and analyzed using the MestReNova (Version 12.0.4) software. The peak heights were used to calculate the change in the intensity of amide backbone peaks of $^{15}$N-αS residues were analyzed manually using MestReNova (Version 12.0.4) software. The reported values are the ratio of the change in the intensity peaks of amide backbone peaks of $^{15}$N-αS residues in the absence and presence of SK-129 at various stoichiometric ratios.

For the lipid membrane conditions in HSQC NMR, a fresh solution of $^{15}$N-αS was prepared at a concentration of 70 μM (350 μL) in the presence of 1 mM DOPS (100 nm, LUVs) for 2D HSQC NMR. All the conditions were exactly similar to HSQC conditions used earlier, including buffer, temperature, and HSQC NMR parameters. The peak heights were used to calculate the change in the intensity of amide backbone peaks of $^{15}$N-αS residues were analyzed manually using MestReNova (Version 12.0.4) software.

To check the effect of SK-129 on the lipid membrane-bound $^{15}$N-αS, a complex was formed between 70 μM $^{15}$N-αS and 1 mM DOPS (100 nm, LUVs). To this solution, SK-129 (70 μM) was added (10 mM stock solution in DMSO) and the solution was mixed gently before running the HSQC NMR experiment. All the conditions were exactly similar to the HSQC conditions used earlier, including buffer, temperature, and HSQC NMR parameters.

All NMR experiments were conducted one time and the NMR experiments between αS and SK-129 were conducted at different stoichiometric ratios and a consistent trend of the chemical shift volume change in the amide peaks was observed as a function of stoichiometric ratios (of αS and SK-129), which supports the reproducibility of the data.

**Culture method for C. elegans**. The PD strain (NL5901), the control strain (N2, Bristol), and *Escherichia coli (E. coli)* OP50 were obtained from the Caenorhabditis Genomics Center (CGC, Minneapolis, MN). Standard conditions were used to maintain the worms at ~23 °C on nematode growth media (NGM) agar plates (60 mm), including *E. coli* OP50 (a uracil requiring mutant of *E.coli* with 0.5 optical density, O.D.) used as a food source for *C elegans*[68]. Stocks of N2 and NL5901 strains of *C. elegans* were cultured and maintained in accordance with the previous protocol[69]. NGM media, M9 buffer, and OP50 food for *C elegans* were prepared according to previous protocols[68,69]. M9 buffer was used for the preparation of liquid media for this experiment. The buffer was prepared by dissolving 3 g of potassium dihydrogen phosphate (KH$_2$PO$_4$), 6 g of sodium hydrogen phosphate (Na$_2$HPO$_4$), and 5 g of sodium chloride (NaCl) in 1 L of Milli-Q water. The M9 buffer was autoclaved for 47 min and subsequently, 1 mL of 1 M magnesium sulfate (MgSO$_4$) was added to the buffer and it was stored at RT for making the liquid media.

**Antagonist effect of SK-129 on the PD phenotypes in the NL5901 strain**. The assay was carried out based on a previous protocol[57] with slight modifications. On day one, N2 and NL5901 strains were synchronized through the bleaching process involving egglay and the incubation of the eggs at an ambient temperature (~23 °C) on a solution of 3 mL NGM media in 60 mm culture plates (CytoOne, USA Scientific, Ocala, FL). The plates were seeded with 300 μL OP50 at an O.D. of 0.5 for 30 h. Simultaneously, a solution of 15 μM SK-129 (Stock solution concentration = 10 mM in DMSO) was made in M9 buffer and added to two sterilized 35 mm NGM solid media plates (Fisher Scientific, Pittsburgh, PA). The plates were already containing 2 mL of NGM solid media and 75 μM Fluorodeoxyuridine (FUDR) to prevent offspring in the worms[69]. On day two, M9 buffer was used to transfer the worms from 60 mm NGM plates into 35 mm FUDR and SK-129 containing solid media plates (Fisher Scientific, Pittsburgh, PA) and incubated up to day three on these plates. On day four, liquid media was prepared according to the previous protocol[69] with some modifications. On day four, liquid media was prepared according to the previous protocol[69,70] with some modifications including 67.28% (v/v) of M9 buffer, 0.018% (75 μM) of FUDR solution (v/v), 0.1% of 1 M magnesium sulfate (v/v), 0.1% of 1 M calcium chloride (v/v), 2.5% of 1 M potassium phosphate solution (pH 6, v/v), and 30% of 0.5 OD$_{600nm}$ OP50 (v/v). A total of 100 worms were transferred using a worm pick into each liquid media plate, including N2, NL5901 in the absence and presence of SK-129. For each experiment, two plates were used for each condition. For the NL5901 strain, the solution conditions were exactly the same in the absence and presence of SK-129, except that SK-129 was present in one condition (15 μM SK-129, stock solution concentration = 10 mM in DMSO). On day five, motility assay was conducted for worms under different conditions in duplicate using a WMicroTracker Arena plate reader (Phylumtech, Argentina) at ~23 °C for 1 h per day over a period of 12 days. Before each run, the plates were gently tapped for 30 sec to enable the worms to become active in the liquid media. A total of 20 activity scores were collected per strain per condition over 1 h on each day. For each condition, at least two technical replicates and three biological replicates were used[12]. The data to extract the mean and s.e.m. ($n = 3$ independent experiments) was processed using GraphPad Prism (Version 9.3.1) software.

**Confocal imaging of C elegans (NL5901)**. The worms from the NL5901 strain were treated in the absence and presence of SK-129 (15 μM SK-129, stock solution concentration = 10 mM in DMSO) as described in the paralysis experiment. The worms (days of adulthood = 8 days) were placed in a 20 mM solution of sodium azide for 5 min. Simultaneously, a glass cover slip was prepared with a drop of 2% agarose pad on it. Subsequently, the worms were mounted with a coverslip on a 2% agarose pad. The worms were visualized on an Olympus Fluoview FV3000 confocal/2-photon microscope, using a 20× (or 40×) Plan-Apo/1.3 NA objective with DIC capability. The confocal images of the worms were processed using the OlympusViewer in ImageJ processing software. The system acquires a series of frames at specific Z-axis position (focal plane) using a Z-axis motor device. For each experiment, at least 15 worms were used, and the inclusions were counted manually, and each condition consisted of at least four independent experiments.

**Reporting summary**. Further information on research design is available in the Nature Research Reporting Summary linked to this article.

## Data availability

All the datasets generated and analyzed during the current study are also available from the corresponding author. Source data is available for Figs. 1–6 and Supplementary Figs. 1–4, 9–13, 16–18, and 23–28 in the associated source data file. Source data are provided with this paper.

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

## Acknowledgements

The authors like to thank the department of chemistry and biochemistry, The Knoebel Institute for Healthy Aging, and the University of Denver for the startup up funds. The author also thanks the PinS program (University of Denver) for awarding summer undergraduate fellowships to T.C.F., C.M.D., and M.M.B. We also like to thank Prof. Marc Diamond's lab for the wonderful gift of the HEK cells that stably express YFP-labeled WT αS (αS-YFP) and a familial mutant, A53T (αS-$_{A53T}$-YFP). We would also like to thank the American Parkinson Disease Association Grant (2021) to support the research conducted in this manuscript. We would also like to thank the Parkinson's Foundation for the summer student fellowship to C.M.D. We sincerely thank The Fitch lab at NYU, Department of Biology, especially Prof. David Fitch and Dr. Karin Kiontke, for providing training to Sunil Kumar, which helped him in establishing *C elegans* based PD system in his lab. We also thank Prof. Lotta Granholm-Bentley for the comments and proofreading of this manuscript. We also like to acknowledge the financial support from the Movement Disorder Foundation (MDF).

## Author contributions

S.K. designed and conceived the project with assistance from J.A. The synthesis of OQs and their derivatives were carried out by S.K. and J.A. The biophysical study was carried out by J.A. with some assistance from S.K. The NMR study was carried out by J.A. with some assistance from S.K. The design, expression, and purification of the αS mutants and the WT αS was carried out by J.A. The biophysical characterization of αS mutants was carried out by J.A. The SH-SY5Y cell toxicity assays were carried out by T.C.F. The HEK cell-based cytotoxicity and confocal microscopy imaging were carried out by C.M.D. with initial assistance from T.C.F. The extraction of the αS seeds from the post mortem PD brain was carried out by J.A. with the help from T.C.F. The *C elegans*-based in vivo experiments to monitor the locomotion were carried out by J.A.J. The confocal microscopy imaging experiments with *C elegans* and HEK cells were conducted by S.K. and C.M.D. The primary hippocampal neuron experiments were carried out by J.A. with the help of C.Z. from Y.Q.'s lab. The toxicity assays for primary hippocampal neurons were carried out by J.A. and T.C.F. The confocal microscopy imaging and immunochemistry experiments with the primary hippocampal neurons were carried out by J.A. with assistance from S.K. The flow cytometry experiments were conducted by S.K. and A.S., with assistance from S.H. The paper was written by S.K. with assistance from J.A., with editing from S.H.

## Competing interests

The authors declare no competing interests.
