## [Peer Review File · Nature Communications]

Reviewers' Comments:

Reviewer #1:

Remarks to the Author:

In "Foldamers reveal and validate novel therapeutic targets associated with toxic α -synuclein self assembly", Ahmed J and co-authors described how a oligoquinoline-based foldamer (SK-129) is capable of inhibiting the aggregation and seeding of α S in biophysical assays in vitro, in cells and in vivo. In particular, they show that SK-129 binds to α S and stabilise the monomer in an α -helical conformation and they show using NMR that the SK-129 binding to α S involves amino acids in the N-terminus and NAC region of the protein. The authors proposed that SK-129 compete with DOPS LUVs for the binding to α S and it would strengthen this hypothesis if this interaction would be characterised further. For ex, do SK-129 molecules form micelles under these conditions? What is the KD for the binding α S-DOPS LUVs? Using additional biophysical assays, the authors show that SK-129 inhibits the aggregation of α S. It is difficult for the reader to assess the reproducibility of the aggregation assay as only one curve is shown for α S +/- SK-129. Could the authors add the change in the ThT fluorescence with time for all the replicates for the different conditions? The authors also conclude from the fact that the ThT fluorescence is not increased in the presence of SK-129 that the molecule inhibits α S aggregation. Did the authors checked that SK-129 did not quench the ThT fluorescence? Could the measurement of the protein at the end of the incubation be measured and compared to the initial concentration of the protein to conclude that none of it indeed aggregated in the presence of SK-129?

The authors also show that SK-129 can prevent the aggregation and the spreading of α S in cells and in vivo.

The manuscript is overall well written and the work described in this study is a very good contribution to the field. I therefore recommend this paper for publication once the comments, described above, have been addressed.

Reviewer #2:

Remarks to the Author:

The manuscript entitled "Foldamers Reveal and Validate Novel Therapeutic Targets Associated with Toxic α -Synuclein Self-Assembly" by Sunil Kumar and co-workers in my view should impact contemporary efforts in biomimetic and targeting proteins. The combination of well designated foldamers, together with selected proteins is convincing. The manuscript itself is well-crafted and remarkably well-written in clear and concise fashion. The evidence supporting the central claims of the paper is extensively discussed with the aid of carefully designed figures, NMR spectra, and modeling studies, and biological experiments. Methods give evidence for the proposed claims, thus they are solid and convincing as well. I recommend publication in "nature communication" virtually "as is".

In the introduction, it should be cited Angew. Chem. Int. Ed. 2018, 57, 10217 –10220., where novel foldamers are able to induce α S controllable aggregation.

Reviewer #3:

Remarks to the Author:

In this study Ahmed and colleagues describe the development of a new therapeutic compound for inhibiting the aggregation of α -synuclein, a protein implicated in the disease process of multiple synucleinopathies – Parkinson's, dementia with Lewy Bodies and multiple system atrophy. This new compound, SK-129, is a small molecule that efficiently and directly binds α Syn and prevents its aggregation into high molecular weight assemblies, that are considered toxic and cause pathology. The study is interesting, and the way how SK-129 binds and blocks aggregation shows promise.

SK-129 effectively inhibits the aggregation of α Syn, via direct binding, and this is shown in multiple assays. The mechanism by which this happens is shown and for this I only have minor

comments (see below). In several experiments the authors also examine the spreading or propagation features of aSyn and mention that SK-129 can also block the spread of pathology. These claims, however, are overstated and even misleading. What is clear is that SK-129 can inhibit the aggregation of aSyn in cellular models and *C. elegans* and although it is mentioned several times that SK-129 inhibits the spread of aSyn pathology or its prion-like features, these claims are not supported by the experiments:

1. The authors suggest that via a PMCA assay the spread of aSyn seeds can be examined or that SK-129 can influence prion-like propagation. With PMCA assay it is possible to study the seeding of aSyn it but does not give any information on the actual spread or progression of pathology. If the authors want to study the propagation of aSyn, they must include an animal propagation study, where the cell to cell spreading of seeds and pathology can be examined.

2. The authors state that "Using primary hippocampal neurons, an aS aggregation-based seeding model has been recently developed that recapitulates the key events of aggregation, seeding, and maturation of inclusions that mimic the bona fide LB". Although primary cultures can generate inclusions, they are far from true LBs. Several issues also remain with the primary hippocampal neurons. In figure 4m, it is not clear where the inclusions really are. It seems that the PD fibrils cause diffuse distribution/staining of Pser129, which is unusual, and not reminiscent of most seeding assay with recombinant or patient-derived fibrils. P62 also does not colocalize into the typical aggregates that are usually seen for these inclusions. Treatment of this condition with SK-129 furthermore results in neuritic staining with Pser129. Is there a redistribution of pathology from diffuse to neuritic pathology upon treatment? Finally, the authors overstate the effect of this assay as strictly speaking it is not possible to test the spread of pathology in primary culture since the fibrils are added to the medium of the entire culture. Spreading could be assessed via microfluidic chambers or other systems.

3. The *C. elegans* model also does not show any features of disease spreading or propagation of aSyn pathology. Treatment with SK-129 shows a significant reduction in aSyn positive inclusions, but these inclusions are the result of the aSyn overexpression of the model itself. It is not shown that these inclusions are the result of a spreading or propagation effect.

In general, the effects of SK-129 on aSyn are interesting, and it is likely that SK-129 will inhibit the spread of pathology (which would be interesting to study in a follow up study) but the exact effect or the mechanism by how the foldamer works in terms of biology and pathogenesis is incorrectly/overstated and needs to be thoroughly revised. This also needs to be corrected in the discussion.

Minor:

Description of conditions on x-axis is missing of several of the graphs in figure 1

Since SK-129 competes with aSyn for the membrane, it would be good to know if SK-129 does also not inhibit the native function of aSyn, resulting in a loss of function. The authors should at least discuss this.

The amplification cycles with the kinetics of seeding of the PMCA assay from patient and control brain is not shown (or I could not find it). It is not possible to judge the kinetics and the quality of the PMCA without the actual data.

What is exactly shown in figure 4d and 4e? Are these TEM images of assemblies after extraction (and how were they then extracted and purified) or after amplification?

The discussion is rather limited, and I would like to read more about what can be expected for distribution, BBB permeability, stability in vivo and its usefulness as a drug? Will there be any follow up experiments examining these effects in vivo?

RESPONSE TO REVIEWER COMMENTS: (MANUSCRIPT: NCOMMS-21-19623-T)

Reviewer #1 (Remarks to the Author):

In "Foldamers reveal and validate novel therapeutic targets associated with toxic α -synuclein self assembly", Ahmed J and co-authors described how a oliguinoline-based foldamer (SK-129) is capable of inhibiting the aggregation and seeding of α S in biophysical assays in vitro, in cells and in vivo.

Comment:

In particular, they show that SK-129 binds to α S and stabilise the monomer in an α -helical conformation and they show using NMR that the SK-129 binding to α S involves amino acids in the N-terminus and NAC region of the protein. The authors proposed that SK-129 compete with DOPS LUVs for the binding to α S and it would strengthen this hypothesis if this interaction would be characterised further. For ex, do SK-129 molecules form micelles under these conditions?

Address to the comment:

We agree with the reviewer's comment about the further characterization of the binding of SK-129 to Synuclein in the presence of LUVs. First, we have used TEM to show that SK-129 does not form micelles under these conditions. We have included the TEM image of SK-129 in the supporting information as **Supplementary Fig. 15** showing no micelle formation.

Supplementary Fig. 15. The negatively stained-TEM images at 400 nm (a) and 50 (b) scales of the aggregation of 35 μ M α S in the presence of LUVs (875 μ M, 100 nm, DOPS) in 20 mM NaCl, 20 mM NaPi, pH 6.5 in the presence of SK-129 at an equimolar ratio after seven days under matched .

Comment:

What is the KD for the binding α S-DOPS LUVs?

Address to the comment:

We utilized circular dichroism technique to determine the K_d between α S and LUVs (DOPS), similar to previously published work^{38,42}. A titration was carried out between 50 μ M α S and an increasing conc. of DOPS, which yielded a $K_d > 5 \mu$ M (data not shown). The K_d between α S and DOPS was higher than the K_d between SK-129 and α S (0.70 μ M), which indicates that the binding affinity of SK-129 is much

higher for aS than the binding affinity of LUVs for aS. The experiment suggests that aS would favor binding to SK-129 than DOPS, and our CD data in the main manuscript support this claim. In the main manuscript, the data show the partial displacement of aS from the aS+DOPS complex in the presence of SK-129, indicated by a decrease in the CD signal of the aS+DOPS complex.

A follow up detailed study is underway to characterize the binding interaction between aS and SK-129 in the presence of lipid membranes using various biophysical, cellular, and in vivo assays. The CD based titration study to determine the K_d between aS and DOPS is a part of the follow up study and we respectfully request to keep that data for the follow up manuscript. The study will also test the effect of SK-129 on the native function of aS, which is facilitated by the membrane bound state of aS. We believe that this study is beyond the scope of this current manuscript and we respectfully request that we are allowed to present that study in a separate manuscript.

For the current manuscript, using the results from the CD titration and the NMR-based study from the main manuscript, we have proposed the mechanism of the binding interaction of SK-129 with aS in the presence of LUVs. As suggested by reviewer 3 as well, we have added a paragraph that describes the potential mechanism of the binding interaction of SK-129 with aS under the lipid membrane conditions.

We have added the following writeup to the main manuscript, and it reads as follows:

“However, there was a decrease in the CD intensity of α -helix upon the addition of SK-129, which suggests that SK-129 might be competing against the lipid membrane for aS (Supplementary Fig. 14b). We utilized a CD to determine the K_d between α S and LUVs (DPS, 100 nm), similar to previously published work^{38,42}. A CD titration was carried out between 35 μ M α S and an increasing conc. of DOPS (molar ratio, 0.5:80, α S:DOPS), which yielded a $K_d > 5 \mu$ M (data not shown). The K_d between α S and DOPS was higher than the K_d between SK-129 and α S ($0.72 \pm 0.06 \mu$ M), which indicates that the binding affinity of SK-129 is higher than DOPS for α S. Consequently, α S should favor binding to SK-129 than DOPS, when SK-129 is added to the complex of α S+DOPS. Our CD data support this claim as the addition of SK-129 to the α S+DOPS complex resulted in a decrease in the CD signal intensity of the DOPS bound α S. Under these conditions, SK-129 does not form any micelle structures, as confirmed with the TEM images (Supplementary Fig. 15). Overall, the data suggest that SK-129 specifically interacts with α S and competes with the LUVs for α S and inhibit the aggregation of lipid catalyzed aggregation”.

Comment:

Using additional biophysical assays, the authors show that SK-129 inhibits the aggregation of aS. It is difficult for the reader to assess the reproducibility of the aggregation assay as only one curve is shown for aS +/- SK-129. Could the authors add the change in the ThT fluorescence with time for all the replicates for the different conditions?

Address to the comment:

As per the reviewer's concern, we have now added the average and the error bars for the change in the ThT fluorescence intensity for the aggregation curve of aS \pm SK-129 (Main manuscript, Fig. 1d). In addition, we have included the average and error bars for the change in the ThT fluorescence intensity for the aggregation curve of aS variants (A53T and A30P) \pm SK-129 (Supplementary Fig. 1). Also, we have included the average and error bars for the change in the ThT fluorescence intensity for the aggregation curves of aS mutants \pm SK-129 (Fig. 3f-j) and aS seed catalyzed aggregation of aS mutants \pm SK-129 (Fig. 3u-y).

Main manuscript, Fig. 1d

Supplementary Fig. 1

Main manuscript, Fig. 3f-j

Main manuscript, Fig. 3u-y

Comment:

The authors also conclude from the fact that the ThT fluorescence is not increased in the presence of SK-129 that the molecule inhibits aS aggregation. Did the authors checked that SK-129 did not quench the ThT fluorescence?

Address to the comment:

We appreciate the comment from the reviewer about checking the fluorescence quenching ability of SK-129 for ThT dye in the aggregation assay of aS. This experiment is important to ensure that SK-129 is inhibiting aS aggregation and decreasing the ThT fluorescence signal instead of quenching the ThT fluorescence intensity. We have confirmed that SK-129 does not have any significant quenching effect on ThT fluorescence intensity (Supplementary Fig. 3) under the matched conditions of the ThT-based aggregation assay. To test the effect of SK-129 on ThT fluorescence signal, we used ThT dye at a final concentration of 50 μM and then added a 100 μM SK-129 solution in the aggregation assay buffer (20 mM NaCl, 20 mM NaPi, pH 6.5). We checked the final ThT fluorescence of both the solutions, including ThT control and the ThT+SK-129 solution after 4 days. We did not observe any significant quenching effect of SK-129 on ThT fluorescence intensity. In addition, we have shown using various techniques, including TEM images (Supplementary Fig. 2c), SDS-PAGE (Supplementary Fig. 4a,c), and gel shift assay (Supplementary Fig. 5a) that SK-129 potently inhibits the aggregation of aS. The figure has been added to the supporting information and a sentence to the main manuscript:

“Under matched conditions, we did not observe any significant quenching of the ThT fluorescence signal by SK-129 (Supplementary Fig. 3)”.

Supplementary Fig 3. The graphical representation of the fluorescence intensity of 50 μM ThT-dye in the absence (control) and presence (SK-129) of 100 μM SK-129 in the aggregation buffer (20 mM NaCl, 20 mM NaPi, pH 6.5). The fluorescence intensity of the samples was measured after incubation for four days. The experiment was conducted three times and the reported change in the ThT intensity is an average of three separate experiments. The reported error bars are the s.d.'s for three sets of experiments.

Comment:

Could the measurement of the protein at the end of the incubation be measured and compared to the initial concentration of the protein to conclude that none of it indeed aggregated in the presence of SK-129?

Address to the comment:

We agree with the reviewer's comment that we need another experiment that could demonstrate that SK-129 is keeping αS in its monomeric form. We used SDS-PAGE gel shift assay to confirm the antagonist activity of SK-129 against αS aggregation (Supplementary Fig. 4). A solution of 100 μM αS was aggregated for four days in the aggregation buffer (1 \times PBS buffer) in the absence and presence of SK-129 at an equimolar ratio. Subsequently, the αS solutions were centrifuged to separate αS aggregates from the soluble αS . Afterward, we boiled the samples to disassemble αS aggregates and examined them using SDS-PAGE (Supplementary Fig. 4). In addition, we also quantified SDS-PAGE gel band intensities using ImageJ software. In the absence of SK-129, αS was predominantly detected in the insoluble fraction (αS aggregates) (Supplementary Fig. 4a,b). In marked contrast, in the presence of SK-129, αS was predominantly detected in the soluble fraction (αS monomer) (Supplementary Fig. 4a,c). These results clearly demonstrate that SK-129 is a potent inhibitor of αS aggregation. The figure (Supplementary Fig. 4) and the writeup have been included in the manuscript. Also, the procedure to conduct this experiment has been added to the material and methods section in the manuscript.

Writeup in the main manuscript:

"The antagonist activity of SK-129 for αS aggregation was also analyzed using SDS-PAGE (Supplementary Fig. 4). A solution of 100 μM αS was aggregated for four days in the aggregation buffer (20 mM NaCl, 20 mM NaPi, pH 6.5) in the absence and presence of SK-129 at an equimolar ratio. Subsequently, the αS solutions were centrifuged to separate αS aggregates from the soluble αS . Afterward, the samples were boiled at 95 $^{\circ}\text{C}$ for 5 min. to disassemble αS aggregates and examined them using sodium dodecyl sulphate–polyacrylamide gel electrophoresis (SDS-PAGE) (Supplementary Fig. 4). In addition, we also quantified SDS-PAGE gel band intensities using ImageJ software. In the absence of SK-129, αS was predominantly detected in the insoluble fraction (αS aggregates) (Supplementary Fig. 4a,b). In marked contrast, in the presence of SK-129, αS was predominantly detected in the soluble fraction (αS monomer) (Supplementary Fig. 4a,c). These results clearly demonstrate that SK-129 is a potent inhibitor of αS aggregation".

Supplementary Fig. 4. SDS-PAGE gel analysis of 100 μ M α S aggregation for four days in the absence and presence of SK-129 at an equimolar ratio in the aggregation buffer (20 mM NaCl, 20 mM NaPi, pH 6.5). **a**, The SDS-PAGE gel showing the Coomassie-stained α S, including total protein (1), soluble fraction (2), and insoluble fraction (3). The α S aggregation was tested in the absence and presence of SK-129 at an equimolar ratio. The reference of the masses (in kD) is shown on the left side of the gel. **b**, The statistical analysis of the relative band intensities of the total α S (1), the soluble fraction (2), and the insoluble fraction (3) of α S. **c**, The statistical analysis of various forms of α S in the presence of SK-129 at an equimolar ratio (α S:129, 1:1). The aggregation kinetics of α S in the absence and presence of SK-129 and the gel shift assays were conducted three times and the reported relative band intensities for various α S fractions is an average of three separate experiments. The reported error bars are the s.d.'s for three separate experiments. Statistical significance was analyzed using a one-way analysis of variance (ANOVA) with Tukey's multiple comparison's test. * p <0.05, ** p <0.01, *** p <0.001.

Comment:

The authors also show that SK-129 can prevent the aggregation and the spreading of α S in cells and in vivo. The manuscript is overall well written and the work described in this study is a very good contribution to the field. I therefore recommend this paper for publication once the comments, described above, have been addressed.

Address to the comment:

We appreciate the comment from the reviewer about acknowledging that the manuscript is well written and our work is a very good contribution to the field. We sincerely hope that we have addressed all the concerns raised by the reviewer with due diligence and carefully designed and executed experiments.

Reviewer #2 (Remarks to the Author):

The manuscript entitled “Foldamers Reveal and Validate Novel Therapeutic Targets Associated with Toxic α -Synuclein Self-Assembly” by Sunil Kumar and co-workers in my view should impact contemporary efforts in biomimetic and targeting proteins. The combination of well designated foldamers, together with selected proteins is convincing. The manuscript itself is well-crafted and remarkably well-written in clear and concise fashion. The evidence supporting the central claims of the paper is extensively discussed with the aid of carefully designed figures, NMR spectra, and modeling studies, and biological experiments. Methods give evidence for the proposed claims, thus they are solid and convincing as well. I recommend publication in “nature communication” virtually “as is”. In the introduction, it should be cited *Angew. Chem. Int. Ed.* 2018, 57, 10217 –10220., where novel foldamers are able to induce aS controllable aggregation.

Address to the comment:

We appreciate the comment from the reviewer that the manuscript is remarkably well written in clear and concise fashion and the proposed claims are well supported by the carefully designed and executed experiments. Based on reviewer’s recommendation, we have now added the suggested citation in the introduction as citation no. 27 in the main manuscript. The citation is below:

Marafon, G., Crisma, M., Masato, A., Plotegher, N., Bubacco, L., Moretto, A. Photoresponsive Prion-Mimic Foldamer to Induce Controlled Protein Aggregation. *Angew. Chem. Intl. Ed.* **60**, 5173-5178 (2020).

Reviewer #3 (Remarks to the Author):

Comment:

In this study Ahmed and colleagues describe the development of a new therapeutic compound for inhibiting the aggregation of a-synuclein, a protein implicated in the disease process of multiple synucleinopathies – Parkinson’s, dementia with Lewy Bodies and multiple system atrophy. This new compound, SK-129, is a small molecule that efficiently and directly binds aSyn and prevents its aggregation into high molecular weight assemblies, that are considered toxic and cause pathology. The study is interesting, and the way how SK-129 binds and blocks aggregation shows promise. SK-129 effectively inhibits the aggregation of aSyn, via direct binding, and this is shown in multiple assays. The mechanism by which this happens is shown and for this I only have minor comments (see below).

Address to the comment:

We would like to thank the reviewer for the positive comments about the study and acknowledging that the data supports the identification of a novel small molecule, SK-129, as a potent inhibitor of the aggregation of aSyn, and its importance in several diseases. We also appreciate the reviewer’s comment that we have presented sufficient evidence using multiple assays to support that SK-129 is a potent inhibitor of aSyn aggregation and associated toxicity and pathology.

Comment:

In several experiments the authors also examine the spreading or propagation features of aSyn and mention that SK-129 can also block the spread of pathology. These claims, however, are overstated and even misleading. What is clear is that SK-129 can inhibit the aggregation of aSyn in cellular models and *C. elegans* and although it is mentioned several times that SK-129 inhibits the spread of aSyn pathology or its prion-like features, these claims are not supported by the experiments: 1. The authors suggest that via a PMCA assay the spread of aSyn seeds can be examined or that SK-129 can influence prion-like propagation. With PMCA assay it is possible to study the seeding of aSyn it but does not give any information on the actual spread or progression of pathology. If the authors want to study the propagation of aSyn, they must include an animal propagation study, where the cell to cell spreading of seeds and pathology can be examined.

Address to the comment:

To address this comment, we have revised the whole manuscript based on our experimental data that confirms that SK-129 is a potent inhibitor of “the seed catalyzed aggregation of α -Synuclein” as opposed to “the prion-like spread of α -Synuclein or the propagation or spread of pathology”. The claim has been revised in various sections of the manuscript, including the Introduction, Results, and Discussion.

As suggested by the reviewer (in a later comment) that the effect of SK-129 on the disease spreading and propagation due to aSyn seeds would be interesting to study in a follow up study. We agree with the reviewer and believe that the animal work to study the effect of SK-129 on the disease spreading and propagation due to aSyn seeds is beyond the scope of the present study. A study is underway in our lab to assess the effect of SK-129 on the spreading and propagation of PD phenotypes mediated by aSyn seeds in a well-established mouse model of PD (α S_{A53T} transgenic line M83)^{61,62}. This mouse model has been studied extensively because it mimics the PD pathologies^{61,62}. The study will assess the effect of SK-129 on PD phenotypes mediated by the spread of aSyn seeds. The study will be

published in the near future and we respectfully request that we are allowed to publish this study as a separate manuscript. As suggested by the reviewer in a later comment, we have added a paragraph in the discussion for the future study.

Writeup in the main manuscript (Discussion):

“A PD mouse model-based study is underway to further assess the pharmaceutical properties and the antagonist activity of SK-129 against PD phenotypes. We are using a well-established mouse model of PD (α S_{A53T} transgenic line M83)^{61,62}. This mouse model has been studied extensively because it mimics the PD pathologies^{61,62}. Using this model, we will be able to assess the pharmacokinetics and pharmacodynamics properties of SK-129. Also, the mouse model study will be used to assess the ability of SK-129 to cross the blood brain barrier. The mouse model study will be used to assess the antagonist activity of SK-129 against PD phenotypes. In addition, the PD mouse model will be treated with PD fibrils in the absence and presence of SK-129. The study will be used to assess the effect of SK-129 on the spreading and propagation of PD phenotypes facilitated by PD fibrils”.

Comment:

2. The authors state that “Using primary hippocampal neurons, an α S aggregation-based seeding model has been recently developed that recapitulates the key events of aggregation, seeding, and maturation of inclusions that mimic the bona fide LB”. Although primary cultures can generate inclusions, they are far from true LBs.

Address to the comment:

We agree with reviewer’s comment that the aSyn inclusions in primary cultures neurons are far from true LBs, which is the reason we had mentioned that the inclusions in primary culture neurons mimic various features of LB and they are not the exact LBs. In the agreement with reviewer’s comment, we have changed the terminology in the manuscript from LBs to LB-like features of aSyn inclusions. The similar terminology has been used (LB-like features of aSyn inclusions) in literature for the matured inclusions in primary culture neurons⁵⁰.

Comment:

Several issues also remain with the primary hippocampal neurons. In figure 4m, it is not clear where the inclusions really are. It seems that the PD fibrils cause diffuse distribution/staining of P_{Ser129}, which is unusual, and not reminiscent of most seeding assay with recombinant or patient-derived fibrils.

Address to the comment:

We want to point out that in the presence of PD fibrils, P_{Ser129} (red) colocalizes in the primary culture neurons, most likely in the cytoplasm as suggested by others as well^{51,52}. Our results (Figure 4m, below) from the confocal imaging resembles earlier published results (Ref. 51 in the main manuscript). The published report claims that the aggregates are in the cytoplasm region of neurons. We have added a Figure from the published report (Fig. 5a from Ref. 51) for the comparison. From both figures, including the confocal image from our results (Fig. 4m) and from the published report (Fig. 5a from Ref. 51), we can clearly see that P_{Ser129} colocalizes (red) with P62.

In addition, we have also stained these aggregates with an aggregate staining dye, ThS. The results from various staining agents suggest that P_{Ser129} aggregates colocalize in the primary culture neurons in the presence of PD fibrils. We have clarified this in the main manuscript.

Part of Figure 4m (Main manuscript)

Fig. 5a (from Ref. 51). At 5 DIV, primary hippocampal neurons were transduced with human WT a-syn Pffs as described. At 19 DIV, cells were either fixed for IF or harvested for IB analysis. IF using p62, and p-a-syn antibodies, demonstrating that neuronal a-syn aggregates co-localized with p62.

Comment:

P62 also does not colocalize into the typical aggresomes that are usually seen for these inclusions.

Address to the comment:

We respectfully disagree with the reviewer as there are a number of reports that show that P62 colocalizes into the typical aggresomes of aSyn in many biological systems, including mouse model (Lohman, S. *et al.*, Sacino, A.N. *et al.*), human PD brain (Jarvela, T.S. *et al.*), primary culture neurons^{50,51}, and HEK cells⁵¹. The confocal images (Figures below) in these published reports clearly demonstrate that P62 colocalizes in the typical aggresomes of aSyn inclusions.

Lohmann, S. *et al. Acta Neuropathologica*,138, 515–533 (2019).

Sacino, A.N. *et al. The Journal of Neuroscience*, 34,12368 –12378 (2014).

Fig. 1 (from Ref. 51). IF images of HEK293 aSyn cells 24 h after treatment with preformed fibrils of aSyn and stained with Pser129 antibody (*red*), DAPI (*blue*), or antibodies to p62 (*green*). In preformed fiber treated conditions, aSyn aggregates showed strong colocalization with p62.

Fig. 7 (from Ref. Lohmann, S. et al.). Deposits of Pser129 colocalize with p62 in the brain of diseased TgM83+/- mice. Immunofluorescence staining of tissue sections of the brain stem show that Pser129 (*red*) colocalized with p62 (*green*) in affected neurons of diseased mice.

Fig. 5a (from Ref. 51). At 5 DIV, primary culture neurons were transduced with human WT a-syn preformed fibrils. At 19 DIV, cells were either fixed for IF. IF using p62 and Pser129 antibodies, demonstrating that neuronal aSyn aggregates co-localized with p62.

Clearly, our observation of the colocalization of P62 in the typical aggresome of aSyn inclusions in the presence of PD fibrils corroborates well with these published reports.

Comment:

Treatment of this condition with SK-129 furthermore results in neuritic staining with P_{Ser129}. Is there a redistribution of pathology from diffuse to neuritic pathology upon treatment?

Address to the comment:

We agree with the reviewer's comment that the treatment of primary culture neurons with PD fibrils in the presence of SK-129 results in neuritic staining of P_{Ser129}. We want to emphasize that we observed similar staining of P_{Ser129} in the case of the control conditions (no PD fibrils) as well. One of the likely reasons for this staining is a much longer incubation time for the primary neurons with various conditions. In comparison to the literature reports (maximum time = 21 days)^{51,52}, we incubated the primary culture neurons for a total of 31 days, including a 10 day of incubation period, followed by the addition of PMCA samples (+PD or PD+SK-129) and incubation for another 21 days. The reported total incubation time (in literature) with conditions for the primary culture neurons was much shorter (maximum time = 21 days)^{50,51}; however, we incubated the primary culture neurons for a total of 31 days. The reason for the longer incubation time for the primary neurons in the presence of PD sample was because we did not observe any significant intracellular aggregation and neurotoxicity at shorter incubation times in the presence of the PD sample. The difference in the incubation time (literature vs our experiment) required to induce neurotoxicity in the primary culture neurons was likely due to the difference in the α S fibril polymorphs of our experiment and the literature sample. It has been shown earlier that different α S fibril polymorphs could differ in templating α S aggregation and inducing toxicity⁴⁷⁻⁴⁹. At 31 days of incubation time, we observed both aggregation and significant neurotoxicity in the primary culture neurons in the presence of PD fibrils. The primary culture neurons treated with the PD sample (Fig. 4m) were stained after 31 days and they were stained positive for α S-pS-129 (phosphorylated residue 129 in WT α S) (red color) and ThS (blue color), a dye that specifically binds protein aggregates (Fig. 4m)⁵². In addition, these aggregates were stained positive and colocalized for p62 (autophagosome vesicles) and TOM20 (mitochondria) as well (Fig. 4m) and the aggregates were most likely colocalized in the cytoplasmic region of the neurons as suggested by others as well⁵⁰⁻⁵¹. Our confocal imaging data corroborate well with the earlier published work⁵⁰⁻⁵¹. The data suggest that α S inclusions recruit and sequester various organelles, proteins, and membranous structures, similar to the published work^{50,51,53,54}. The staining profile of the primary neurons was very similar for both the control and PD fibrils+SK-129 conditions. We did not observe any colocalization of α S-pS-129 with p62, TOM20 in both the control and PD+SK-129 conditions (Fig. 4m, +PD+SK-129). Also, no staining of α S-pS-129 was observed with ThS dye for both the control and PD+SK-129 conditions (Fig. 4m). We observed mild staining and diffusion of α S-pS-129 in both the control and PD+SK-129 conditions; however, we did not observe any colocalization of α S-pS-129 with any other biomarker, including ThS dye (Fig. 4m). The partial staining of α S-pS-129 is likely due to the longer incubation time (31 days) for the primary culture neurons in our experimental conditions, which might have contributed to some neurotoxicity and the mild staining of α S-pS-129. We used lactate dehydrogenase (LDH) release assay to determine the neurotoxicity of the primary culture neurons in the presence of PD fibrils (\pm SK-129). We observed very high neurotoxicity (~8-fold higher than control) in primary neurons in the presence of the PD sample. In marked contrast, similar to the control sample, we did not observe any significant neurotoxicity in the presence of PD+SK-129 condition (Fig. 4n).

To further validate these results, we used HEK cells-based model, which expresses endogenous monomeric α S-A53T-YFP. In the presence of PD fibrils, both P62 and α S-pS-129 colocalized in the aggresome of α S inclusions in HEK cells after 24 h (Supplementary Fig. 23). The α S inclusions were colocalized in the cytoplasmic region of the HEK cells as suggested by others as well⁵¹. Our results corroborate well with the earlier published work with HEK cells (Supplementary Fig. 23a,b)⁵¹. In marked contrast, in the presence of PD fibrils+SK-129 condition, we detected a significantly smaller number of colocalization of P62 and α S-pS-129 in the aggresome of α S inclusions (Supplementary Fig. 23a,b). In addition, we carried out the MTT reduction-based cytotoxicity assay for the HEK cells in the presence of PD fibrils. The cell viability of HEK cells decreased to 50.5±5.4% in the presence of PD fibrils (Supplementary Fig. 23c). However, in the presence of PD fibrils+SK-129 condition, the cell viability increased to 85.6±7.8% (Supplementary Fig. 23c). Using primary culture neurons and HEK cells, we have shown that various proteins, including P62 and α S-pS-129 colocalize in the aggresome of α S inclusions and mediate toxicity in the presence of PD fibrils. In the presence of PD+SK-129 condition, we observed a significant decrease in the colocalization of P62 and α S-pS-129 in the aggresome of α S inclusions and rescue of the toxicity in HEK cells and primary culture neurons. We have included those results as Supplementary Figure 23 and the writeup has been included in the main manuscript. Also, we have included the explanation and clarification of the antagonist effect of SK-129 on the primary culture neurons toxicity in the presence of PD fibrils.

We have added the explanation and clarification for the effect of SK-129 on the primary culture neurons toxicity and the results from the HEK cells-based experiment in the main manuscript:

“To further confirm the antagonist activity of SK-129 on the seed catalyzed aggregation of α S in the presence of the PD brain extract, we employed a more physiologically relevant model based on the primary rat hippocampal neurons⁵⁰⁻⁵¹. Using primary hippocampal neurons, an α S aggregation-based seeding model has been recently developed that recapitulates the key events of aggregation, seeding, and maturation of inclusions that partly mimic the features of LB-like structures⁵⁰. We incubated primary culture neurons for a total of 31 days, including a 10 day of incubation period, followed by the addition of PMCA samples (+PD or PD+SK-129) and incubation for another 21 days. The reported total incubation time (in literature) for the primary culture neurons was much shorter (maximum time = 21 days)^{50,51}; however, we incubated the primary culture neurons for a total of 31 days. The reason for the longer incubation time for the primary neurons in the presence of PD sample was because we did not observe any significant intracellular aggregation and neurotoxicity at shorter incubation times in the presence of the PD sample. The difference in the incubation time (literature vs our experiment) required to induce neurotoxicity in the primary culture neurons was likely due to the difference in the α S fibril polymorphs of our experiment and the literature sample. It has been shown earlier that different α S fibril polymorphs could differ in templating α S aggregation and inducing toxicity⁴⁷⁻⁴⁹. At 31 days of incubation time, we observed both aggregation and significant neurotoxicity in the primary culture neurons in the presence of PD fibrils. The primary culture neurons treated with the PD sample (Fig. 4m) were stained after 31 days and they were stained positive for α S-pS-129 (phosphorylated residue 129 in WT α S) (red color) and ThS (blue color), a dye that specifically binds protein aggregates (Fig. 4m)⁵². In addition, these aggregates were stained positive and colocalized for p62 (autophagosome vesicles) and TOM20 (mitochondria) as well (Fig. 4m) and the aggregates were most likely colocalized in the cytoplasmic region of the neurons as suggested by others as well⁵⁰⁻⁵¹. Our confocal imaging data corroborate well with the earlier published work⁵⁰⁻⁵¹. The data suggest that α S inclusions recruit and sequester various organelles, proteins, and membranous structures, similar to the published work^{50,51,53,54}. The staining profile of the primary neurons was very similar for both the control and PD fibrils+SK-129 conditions. We did not observe any colocalization of α S-pS-129 with p62, TOM20 in both the control and PD+SK-129 conditions (Fig. 4m, +PD+SK-129). Also, no staining of α S-pS-129 was observed with ThS dye for both the control and PD+SK-129 conditions (Fig. 4m). We observed mild staining and diffusion of α S-

pS-129 in both the control and PD+SK-129 conditions; however, we did not observe any colocalization of α S-pS-129 with any other biomarker, including ThS dye (Fig. 4m). The partial staining of α S-pS-129 is likely due to the longer incubation time (31 days) for the primary culture neurons in our experimental conditions, which might have contributed to some neurotoxicity and the mild staining of α S-pS-129. We used lactate dehydrogenase (LDH) release assay to determine the neurotoxicity of the primary culture neurons in the presence of PD fibrils (\pm SK-129). We observed very high neurotoxicity (\sim 8-fold higher than control) in primary neurons in the presence of the PD sample. In marked contrast, similar to the control sample, we did not observe any significant neurotoxicity in the presence of PD+SK-129 condition (Fig. 4n).

To further validate these results, we used HEK cells-based model, which expresses endogenous monomeric α S-A53T-YFP. In the presence of PD fibrils, both P62 and α S-pS-129 colocalized in the aggresome of α S inclusions in HEK cells after 24 h (Supplementary Fig. 23). The α S inclusions were colocalized in the cytoplasmic region of the HEK cells as suggested by others as well⁵¹. Our results corroborate well with the earlier published work with HEK cells (Supplementary Fig. 23a,b)⁵¹. In marked contrast, in the presence of PD fibrils+SK-129 condition, we detected a significantly smaller number of colocalization of P62 and α S-pS-129 in the aggresome of α S inclusions (Supplementary Fig. 23a,b). In addition, we carried out the MTT reduction-based cytotoxicity assay for the HEK cells in the presence of PD fibrils. The cell viability of HEK cells decreased to $50.5\pm 5.4\%$ in the presence of PD fibrils (Supplementary Fig. 23c). However, in the presence of PD fibrils+SK-129 condition, the cell viability increased to $85.6\pm 7.8\%$ (Supplementary Fig. 23c). Using primary culture neurons and HEK cells, we have shown that various proteins, including P62 and α S-pS-129 colocalize in the aggresome of α S inclusions and mediate toxicity in the presence of PD fibrils. In the presence of PD+SK-129 condition, we observed a significant decrease in the colocalization of P62 and α S-pS-129 in the aggresome of α S inclusions and rescue of the toxicity in HEK cells and primary culture neurons”.

Supplementary Fig. 23. Effect of SK-129 on the PD fibrils catalyzed aggregation of αS templated by PMCA technique. **a**, The representative confocal images of HEK cells after treatment with PMCA samples from the PD and control brain. The green color images are due to the intracellularly expressed αS_{A53T} -YFP. The αS_{A53T} -YFP inclusions, αS -pS-129, and DAPI are represented by green, red, and blue colors. The αS_{A53T} -YFP inclusions are indicated by white arrows. **b**, The αS_{A53T} -YFP inclusions, P62, and DAPI are represented by green, red, and blue colors. The αS_{A53T} -YFP inclusions are indicated by white arrows. **c**, The statistical analysis of the relative viability of HEK cells treated with the indicated conditions for 24 h determined using the MTT-reduction toxicity assay. The cell viability assays were conducted with at least four biological replicates and four technical replicates for each biological replicate and the reported error bars are the s.d.'s for every experiment. The reported error bars are the s.d.'s for multiple set of experiments conducted on separate occasions. Statistical significance was analyzed using a one-way ANOVA with Tukey's multiple comparison's test. * $p < 0.05$, ** $p < 0.01$, *** $p < 0.001$.

We hope that we have addressed the reviewer's comment by additionally conducting HEK-based experiments to demonstrate that both P62 and P^{Ser129} colocalize with the typical aggregates of aSyn inclusions in the presence of PD fibrils and there was a significant reduction in the toxicity and colocalization of P62 and P^{Ser129} in the presence of SK-129. Also, we hope that we have convinced the reviewer by using the confocal images of the primary culture neurons/HEK cells and the LDH/MTT based toxicity assays that SK-129 was a potent antagonist of the seed catalyzed aggregation of aSyn in the primary culture neurons/HEK cells.

Comment:

Finally, the authors overstate the effect of this assay as strictly speaking it is not possible to test the spread of pathology in primary culture since the fibrils are added to the medium of the entire culture. Spreading could be assessed via microfluidic chambers or other systems.

Address to the comment:

We agree with the reviewer's comment that it is not possible to test the spread of pathology using the primary culture neurons because the PD fibrils were added to the medium. We apologize for overstating the antagonist effect SK-129 on the spread of propagation of the disease using this assay in the current manuscript (primary culture neurons+PD fibrils). We have revised the whole manuscript by changing "the prion like spread" or "the propagation or spread of pathology" effect to the "seed catalyzed aggregation". The changes have been incorporated in the main manuscript, which is also presented below.

Also, an animal study is underway to assess the effect of SK-129 on the spread of pathology and it will be presented in a separate manuscript. We agree with the reviewer who has suggested that the effect of SK-129 on the spread of pathology "**it would be interesting to study in a follow up study**".

Comment:

3. The *C. elegans* model also does not show any features of disease spreading or propagation of aSyn pathology. Treatment with SK-129 shows a significant reduction in aSyn positive inclusions, but these inclusions are the result of the aSyn overexpression of the model itself. It is not shown that these inclusions are the result of a spreading or propagation effect.

Address to the comment:

We agree with reviewer's comment that the *C. elegans* PD model (NL5901) in our study does not show any features of the disease spreading or propagation of aSyn pathology. We apologize for the confusion, but in the main manuscript for the section titled '**Effect of SK-129 on α S aggregation mediated PD phenotypes in an *in vivo* model**', we did not claim that we have utilized *C. elegans* PD model to study the effect of SK-129 on the disease spreading or the propagation of aSyn pathology. In this section, we have demonstrated the antagonist effect of SK-129 on the intracellular aSyn aggregation in a *C. elegans* PD model system (NL5901). The same *C. elegans* model (NL5901) has been used earlier to study the effect of small molecules on the intracellular aSyn aggregation in an *in vivo* model^{44,55}.

Comment:

In general, the effects of SK-129 on aSyn are interesting, and it is likely that SK-129 will inhibit the spread of pathology (**which would be interesting to study in a follow up study**) but the exact effect or the mechanism by how the foldamer works in terms of biology and pathogenesis is

incorrectly/overstated and needs to be thoroughly revised. This also needs to be corrected in the discussion.

Address to the comment:

We thank the reviewer in agreeing with the fact that the effect of SK-129 on aSyn aggregation is interesting and has been confirmed with multiple assays. Also, we agree with reviewer's comment that the effect of SK-129 on the spread of pathology will be interesting to study in a follow up study. A follow up study is underway in our lab to assess the effect of SK-129 on the spreading and propagation of PD phenotypes mediated by aSyn seeds in a well-established mouse model of PD (α S^{A53T} transgenic line M83)^{61,62}. This mouse model has been studied extensively because it mimics the PD pathologies^{61,62}. The study will assess the effect of SK-129 on PD phenotypes mediated by the spread of aSyn seeds. The study will be published in the near future and we respectfully request that we are allowed to publish this study as a separate manuscript.

Based on the reviewer's concern, we have thoroughly revised the manuscript and discussion by correcting the antagonist effect of SK-129, which is on "the seed catalyzed aggregation of aSyn", instead of "the prion like spread of aSyn" or "the propagation or spread of pathology". In the whole manuscript (including discussion), we have changed the terminology from "the prion like spread" or "the propagation or spread of pathology" to the "the seed catalyzed mechanism". We have also corrected the overstated claim in the manuscript that SK-129 was a potent inhibitor of the spread and propagation of the pathology. We have modified it with the claim that SK-129 was a potent inhibitor of seed catalyzed aggregation of aSyn. Below are the changes made in the main manuscript as well as in the discussion.

Introduction of the manuscript (BEFORE):

SK-129 was a potent antagonist of the prion-like spread of α S seeds. The activity of SK-129 against the prion-like spread of α S seeds was confirmed using distinct α S seed polymorphs generated from the recombinant α S and extracted from the substantia nigra of the post mortem brain of PD patient. The activity of SK-129 was also confirmed in a novel intracellular assay for the prion-like spread of α S seeds. Overall, SK-129 interacts at the N-terminus of α S monomer, induces or stabilizes a helical conformation, and modulates both *de novo* aggregation and the prion-like spread of α S seeds.

Introduction of the manuscript (AFTER):

"SK-129 was a potent antagonist of the α S seeds catalyzed aggregation of α S monomer. The activity of SK-129 against the α S seeds catalyzed aggregation was confirmed using distinct α S seed polymorphs generated from the recombinant α S and extracted from the substantia nigra of the post mortem brain of PD patient. The antagonist activity of SK-129 was also confirmed in a novel HEK cell-based intracellular assay for the α S seeds catalyzed aggregation of intracellular monomeric α S. Overall, SK-129 interacts at the N-terminus of α S monomer, induces or stabilizes an aggregation incompetent helical conformation, and modulates both *de novo* aggregation and the α S seeds catalyzed aggregation".

Results in the manuscript (BEFORE):

The seed-catalyzed aggregation of α S conceptually mimics the prion-like spread of α S seeds, a phenomenon in which α S seeds enable the rapid conversion of functional and soluble α S into insoluble fibrils⁴¹⁻⁴⁸. An *in vitro* model of the prion-like spread of α S has been recapitulated in the protein misfolding cyclic amplification (PMCA) technique⁴⁹⁻⁵¹. In the PMCA assay, α S fibers are amplified for five cycles using α S monomer and seeds from the previous cycle. Additionally, α S seed polymorphs from different sources differ in mediating PD phenotypes, which is a consequence of the prion-like

spread through infection pathways⁵¹⁻⁵³. Therefore, we utilized two α S seed polymorphs, including recombinant α S seeds and α S seeds extracted from the substantia nigra of a PD brain and a control brain (post mortem condition) and used them in the PMCA assay (Fig. 4a). The Lewy bodies (LBs)/ α S aggregates in the substantia nigra of the PD brain were confirmed using immunostaining (bluish/black, black arrows) and TEM (Fig. 4c,e). We did not observe LBs in the control brain (Fig. 4b,d).

Results in the manuscript (AFTER):

“The antagonist activity of SK-129 on α S aggregation was also assessed using a protein misfolding cyclic amplification (PMCA) technique. The PMCA technique is used to cyclically amplify the aggregation of proteins from a small quantity and diverse species and it also generates robust seeds via a nucleation dependent polymerization model⁴⁴⁻⁴⁶. In the PMCA assay, α S fibers are amplified for five cycles using α S monomer and seeds from the previous cycle. Additionally, α S seed polymorphs from different sources differ in mediating PD phenotypes, which is a consequence of their spread through various infection pathways⁴⁷⁻⁴⁹. Therefore, we utilized two α S seed polymorphs, including recombinant α S seeds and α S seeds extracted from the substantia nigra of a PD brain and a control brain (post mortem condition) and used them in the PMCA assay (Fig. 4a). The Lewy bodies (LBs)-like structural features in the substantia nigra of the PD brain were confirmed using immunostaining (bluish/black, black arrows) (Fig. 4c). We did not observe any LBs-like structural features in the control brain (Fig. 4b). The aggregates of α S were extracted from the PD brain using a published protocol³⁴. The α S aggregates were extracted and confirmed from the PD brain using TEM (Fig. 4e) and western blot (Fig. 4f). In marked contrast, no aggregates of α S were detected after extraction from the control brain as confirmed by TEM (Fig. 4d). Both samples from the control brain and the PD brain were assessed for their ability to seed and accelerate the aggregation of α S monomer. No noticeable change was observed in the t_{50} or the total ThT intensity for α S aggregation in the presence of the control brain sample. The t_{50} for α S aggregation was 67.4 ± 12.2 and 62.1 ± 3.4 in the absence and presence of the control brain sample, respectively (Supplementary Fig. 21a,b). In marked contrast, both t_{50} (22.7 ± 5.4) and ThT intensity (~ 4 fold) for α S aggregation were significantly enhanced in the presence of the PD brain sample (Supplementary Fig. 21a,b). Clearly, the α S seeds from the PD brain sample template and significantly accelerate α S aggregation via seed catalyzed mechanism”.

Results in the manuscript (BEFORE):

Effect of SK-129 on the prion-like spread of α S seeds in an *ex vivo* PD model. We next investigated the effect of SK-129 at preventing seed-catalyzed aggregation from PD brain samples. The sample (cycle 5) of PD brain extract in the presence of SK-129 was neither aggregated nor PK resistant (Fig. 4a). Also, we observed a lower number of inclusions and improved cell viability for up to four days (Fig. 4j-l). We observed similar behavior of the PMCA sample (cycle 5) from recombinant α S seeds in the absence and presence of SK-129 at an equimolar ratio (Supplementary Fig. 18). The PMCA assay for recombinant protein leads to an abundance of α S fibers (Cycle 5), confirmed with high ThT signal (Supplementary Fig. 18f), TEM image (Supplementary Fig. 18d), PK resistance (Supplementary Fig. 18b), inclusions from confocal imaging (Supplementary Fig. 18g,h,i) and Proteostat dye (Supplementary Fig. 18j), and toxicity (Supplementary Fig. 18k) in HEK cells. In contrast, there was no formation of α S fibers for the PMCA assay (Cycle 5) in the presence of SK-129 as confirmed by ThT intensity (Supplementary Fig. 18f), TEM image (Supplementary Fig. 18e), no PK resistance (Supplementary Fig. 18c), very low number of inclusions from confocal imaging (Supplementary Fig. 18g) and Proteostat dye (Supplementary Fig. 18j) and rescue of toxicity in HEK cells (Supplementary Fig. 18k).

To further confirm the effect of SK-129 on the formation of LBs and on the prion-like spread of α S seeds in the presence of the PD brain extract, we employed a more physiologically relevant model based on the primary rat hippocampal neurons. Using primary hippocampal neurons, an α S aggregation-based

seeding model has been recently developed that recapitulates the key events of aggregation, seeding, and maturation of inclusions that mimic the bona fide LBs⁴⁸. We incubated PMCA samples (cycle 5) of brain extracts in the absence and presence of SK-129 in primary hippocampal neurons for 21 days, a reported timeline to form matured LBs⁴⁸. The neurons treated with PD sample (Fig. 4m) were stained positive for LBs biomarkers, including α S-pS-129 (phosphorylated residue 129) (red color) and ThS, a dye that specifically binds protein aggregates (blue color, Fig. 4m)⁵⁴, p62 (autophagosome vesicles) and TOM20 (mitochondria) (Fig. 4m)⁴⁸. The data suggest that α S inclusions recruit and sequester various organelles, proteins, and membranous structures^{45,48,55-56}. We did not observe any colocalization of α S-pS-129 with p62 and TOM20 when neurons were treated with PD sample in the presence of SK-129 (Fig. 4m, +PD+SK-129). Also, no staining of α S-pS-129 was observed with ThS (Fig. 4m). Using lactate dehydrogenase (LDH) release assay, we observed very high neurotoxicity (~8-fold higher than control) in primary neurons in the presence of PD sample; however, no toxicity was observed in the presence of PD sample treated with SK-129 (Fig. 4n).

Results in the manuscript (AFTER):

“Effect of SK-129 on the seed catalyzed aggregation of α S in an *ex vivo* PD model. We next investigated the effect of SK-129 at preventing seed-catalyzed aggregation from PD brain samples. The sample (cycle 5) of PD brain extract in the presence of SK-129 was neither aggregated nor PK resistant (Fig. 4a). Also, we observed a lower number of inclusions and improved cell viability for up to four days (Fig. 4j-l). We observed similar behavior of the PMCA sample (cycle 5) from recombinant α S seeds in the absence and presence of SK-129 at an equimolar ratio (Supplementary Fig. 22). The PMCA assay for recombinant α S leads to an abundance of α S fibers (Cycle 5), confirmed with high ThT signal (Supplementary Fig. 22f), TEM image (Supplementary Fig. 22d), PK resistance (Supplementary Fig. 22b), high number of inclusions from confocal imaging (Supplementary Fig. 22g,h,i), high ProteoStat dye signal (Supplementary Fig. 22j), and much higher cytotoxicity (Supplementary Fig. 22k) in HEK cells. In contrast, there was no formation of α S fibers for the PMCA assay (Cycle 5) in the presence of SK-129 at an equimolar ratio as confirmed by low ThT intensity (Supplementary Fig. 22f), TEM image (Supplementary Fig. 22e), no PK resistance (Supplementary Fig. 22c), very low number of inclusions from confocal imaging (Supplementary Fig. 22g), low ProteoStat dye signal (Supplementary Fig. 22j) and rescue of cytotoxicity in HEK cells (Supplementary Fig. 22k).

To further confirm the antagonist activity of SK-129 on the seed catalyzed aggregation of α S in the presence of the PD brain extract, we employed a more physiologically relevant model based on the primary rat hippocampal neurons⁵⁰⁻⁵¹. Using primary hippocampal neurons, an α S aggregation-based seeding model has been recently developed that recapitulates the key events of aggregation, seeding, and maturation of inclusions that partly mimic the features of LB-like structures⁵⁰. We incubated primary culture neurons for a total of 31 days, including a 10 day of incubation period, followed by the addition of PMCA samples (+PD or PD+SK-129) and incubation for another 21 days. The reported total incubation time (in literature) for the primary culture neurons was much shorter (maximum time = 21 days)^{50,51}; however, we incubated the primary culture neurons for a total of 31 days. The reason for the longer incubation time for the primary neurons in the presence of PD sample was because we did not observe any significant intracellular aggregation and neurotoxicity at shorter incubation times in the presence of the PD sample. The difference in the incubation time (literature vs our experiment) required to induce neurotoxicity in the primary culture neurons was likely due to the difference in the α S fibril polymorphs of our experiment and the literature sample. It has been shown earlier that different α S fibril polymorphs could differ in templating α S aggregation and inducing toxicity⁴⁷⁻⁴⁹. At 31 days of incubation time, we observed both aggregation and significant neurotoxicity in the primary culture neurons in the presence of PD fibrils. The primary culture neurons treated with the PD sample (Fig. 4m) were stained after 31 days and they were stained positive for α S-pS-129 (phosphorylated residue 129 in WT α S) (red color) and ThS (blue color), a dye that specifically binds protein aggregates (Fig. 4m)⁵². In addition, these aggregates were stained positive and colocalized for p62 (autophagosome vesicles) and TOM20 (mitochondria) as well (Fig. 4m) and the aggregates were most likely colocalized in the cytoplasmic

region of the neurons as suggested by others as well⁵⁰⁻⁵¹. The data corroborate well with the earlier published work⁵⁰⁻⁵¹. The data suggest that α S inclusions recruit and sequester various organelles, proteins, and membranous structures, similar to the published work^{50,51,53,54}. The staining profile of the primary neurons was very similar for both the control and PD fibrils+SK-129 conditions. We did not observe any colocalization of α S-pS-129 with p62, TOM20 in both the control and PD+SK-129 conditions (Fig. 4m, +PD+SK-129). Also, no staining of α S-pS-129 was observed with ThS dye for both the control and PD+SK-129 conditions (Fig. 4m). We observed mild staining and diffusion of α S-pS-129 in both the control and PD+SK-129 conditions; however, we did not observe any colocalization of α S-pS-129 with any other biomarker, including ThS dye (Fig. 4m). The partial staining of α S-pS-129 is likely due to the longer incubation time (31 days) for the primary culture neurons in our experimental conditions, which might have contributed to some neurotoxicity and the mild staining of α S-pS-129. We used lactate dehydrogenase (LDH) release assay to determine the neurotoxicity of the primary culture neurons in the presence of PD fibrils (\pm SK-129). We observed very high neurotoxicity (\sim 8-fold higher than control) in primary neurons in the presence of the PD sample. In marked contrast, similar to the control sample, we did not observe any significant neurotoxicity in the presence of PD+SK-129 condition (Fig. 4n)".

Results in the manuscript (BEFORE):

Effect of SK-129 on the intracellular prion-like spread of α S seeds. SK-129 was potent in inhibiting *in vitro* prion-like spread of α S seeds. We developed a novel assay to test the antagonist activity of SK-129 in an intracellular prion-like spread model of α S using HEK cells. First, we assessed the total time required by α S seeds for the internalization into HEK cells. The HEK cells (expressing α S-A53T-YFP) were exposed to α S seeds (0.125 μ M) extracted from PD brain for various time points (0.5, 4, 8, 12, and 24 h), washed the cells, and incubated for a total of 24 h. The formation of α S inclusions was noticeable within 4 h treatment of cells with α S seeds and inclusions were comparable (\sim 20 inclusions/100 cells) in the case of 8,12, and 24 h treatment of cells (Fig. 5h,i), which suggests that α S seeds were completely internalized in cells within 8 h. The antagonist activity of SK-129 against the intracellular prion-like spread was measured using the HEK cells treated with α S seeds for 8 h. SK-129 (10 μ M) was added to HEK cells that were already treated with α S seeds (0.125 μ M) for 8 h, followed by the incubation for an additional 16 h (total 24 h) (Fig. 5j). There was an abundance of inclusions in the absence of SK-129 after 24 h (\sim 19 inclusions/100 cells, Fig. 5i); however, a low number of inclusions (\sim 1-2 inclusions/100 cells) were observed in the presence of SK-129 (Fig. 5j). We also observed a gradual increase in inclusions and cytotoxicity in the presence of α S seeds for up to 4 days. However, in the presence of SK-129, a low number of inclusions was observed for up to 4 days (2-3 inclusions/100 cells, Fig. 5j), and the cell viability was improved significantly up to four days (Fig. 5k). Therefore, we conclude that SK-129 is a potent inhibitor of the intracellular *de novo* α S aggregation and the prion-like spread of α S seeds.

Results in the manuscript (AFTER):

“The Antagonist Effect of SK-129 on the intracellular seed catalyzed aggregation of α S. SK-129 was very potent antagonist of *in vitro* seed catalyzed aggregation of α S, both in cellular and primary culture neuronal models. However, in these models (cellular and neuronal), the solutions of α S aggregates (\pm SK-129) were prepared extracellular and then introduced to the cells or neurons to determine their ability to template the monomeric α S. Here, we aim to determine the antagonist activity of SK-129 against the seed catalyzed aggregation of α S in a novel intracellular assay using HEK cells. In this novel assay, the seeds of α S will be introduced to the HEK cells, followed by the introduction of SK-129 to the cells. This assay will test the antagonist activity of SK-129 against the seed catalyzed aggregation of α S in an intracellular manner. To develop this assay, first, we assessed the total time required by α S seeds for the internalization into HEK cells. The HEK cells (expressing α S-A53T-YFP) were exposed to α S seeds (0.125 μ M) extracted from PD brain for various time points (0.5, 4, 8, 12,

and 24 h), washed the cells, and incubated for a total of 24 h. The formation of α S inclusions was noticeable within 4 h of the treatment of cells with α S seeds. The number of inclusions were comparable (~20 inclusions/100 cells) in the case of 8,12, and 24 h treatment of cells (Fig. 5h,i), which suggests that α S seeds were completely internalized in cells within 8 h. The antagonist activity of SK-129 against the intracellular seed catalyzed aggregation of α S was measured using the HEK cells treated with α S seeds for 8 h. A solution of SK-129 (10 μ M) was added to HEK cells that were already treated with α S seeds (0.125 μ M) for 8 h, followed by the incubation for an additional 16 h (total 24 h) (Fig. 5j). There was an abundance of inclusions in the absence of SK-129 after 24 h (~19 inclusions/100 cells, Fig. 5i); however, a low number of inclusions (~1-2 inclusions/100 cells) were observed in the presence of SK-129 (Fig. 5j). We also observed a gradual increase in inclusions in the presence of α S seeds for up to 4 days (Fig. 5j). However, in the presence of SK-129, a low number of inclusions was observed for up to 4 days (2-3 inclusions/100 cells, Fig. 5j). In addition, we also observed a gradual decrease in the cell viability of HEK cells treated with PD sample from day one to four (Fig. 4k). In marked contrast, the cell viability was significantly higher upto four days in the presence of SK-129 (Fig. 5k). The data clearly suggest that SK-129 is a potent antagonist of both intracellular *de novo* α S aggregation and the seed catalyzed aggregation of α S”.

Results in the manuscript (BEFORE):

We demonstrated that α S sequences that initiate *de novo* α S aggregation are important for the seed-catalyzed aggregation, which mimics the prion-like spread of α S. The prion-like spread requires the interaction of α S fibers with α S monomers to accelerate and propagate the aggregation. We surmise that SK-129 modulates α S monomer into fiber-incompetent conformation, which consequently prevents the prion-like spread of α S. We also developed a novel intracellular prion-like spread model of α S seeds (extracted from a PD brain). SK-129 was very effective in inhibiting the intracellular prion-like spread of α S seeds.

Results in the manuscript (AFTER):

“We demonstrated that the binding sites of SK-129 (α S sequences) that initiate the *de novo* α S aggregation are also important for the seed-catalyzed aggregation of α S. The seed catalyzed aggregation requires the interaction of α S fibers with α S monomers to accelerate the aggregation. We have shown that SK-129 inhibits the seed catalyzed aggregation of α S. We surmise that the mode of action for SK-129 is a consequence of the interaction of SK-129 with the monomeric α S (towards N-terminal) and the conversion of the latter into a fiber-incompetent conformation, which consequently inhibits the seed catalyzed aggregation of α S. SK-129 was also a potent antagonist of *de novo* aggregation of α S (*C elegans* PD model) and seed catalyzed aggregation of α S (HEK cells) in the intracellular models. Based on our data, we propose that SK-129 permeates the membrane, interacts with the intracellular monomeric α S, and modulates both the *de novo* aggregation of α S (*C elegans*) and the seed catalyzed aggregation of α S (HEK cells). A similar mode of action has been displayed by affibodies^{58,59}, which interact with monomeric α S and modulate it into a β -hairpin conformation. Similar to SK-129, the affibodies were potent inhibitor of both the *de novo* aggregation of α S and the seed catalyzed aggregation of α S^{58,59}”.

Comment: Minor:

Description of conditions on x-axis is missing of several of the graphs in figure 1

Address to the comment:

We apologize for the lack of the description of the conditions for various graphs in Figure 1. We have modified Figure 1 in the main manuscript and have added more description of various graphs in Figure 1. Also, we have added more information about the graphs in the legends of Figure 1.

Fig. 1. Characterization of the antagonist activity of SK-129 against α S aggregation. **a**, The generic chemical structure of the OQ with R_i and R_j are the side chain surface functionalities. The side and top view of the crystal structure of OQs and the surface functionalities are represented by arrows. The OQs with the indicated side chains (R_i and R_j) were used in the study. **c**, Chemical structure of SK-129 and the four side chains were indicated from 1 to 4. **d**, The average of ThT-dye fluorescence-based aggregation profile of 100 μ M α S in the absence and presence of SK-129 at the indicated molar ratios. **e**, The chemical structures of the side chains at position 1 and 3 of various analogs of SK-129. **f**, The antagonist activities of the analogs (100 μ M) of SK-129 against 100 μ M α S aggregation. **g**, The fit for the FP titration curve to determine the binding affinity between 10 μ M SK-129_F and α S. The chemical structure of SK-129_F is shown as well. **h**, The statistical analysis of the relative viability of SH-SY5Y cells when treated with the aggregated solution of 10 μ M α S in the absence and presence of SK-129 at the indicated molar ratios. **i**, Confocal images of HEK cells treated with the aggregated solution of 7 μ M α S in the absence and presence of SK-129 at an equimolar ratio. Inclusions of α S_{A53T}-YFP = white arrows, Hoechst (blue), merge = Hoechst and α S_{A53T}-YFP. **j**, The flow cytometry based analysis of HEK cells treated with the aggregated solution of 7 μ M α S in the absence and presence of SK-129 at an equimolar ratio. The x-axis represents α S_{A53T}-YFP aggregates containing cells that are stained with Proteostat dye ($\lambda = 640$ nm) and the y-axis represents the total number of cells with YFP ($\lambda = 490$ nm). **k**, Columns A and B represent the relative % of HEK cells without and with α S_{A53T}-YFP aggregates, respectively. **l**, The number of α S_{A53T}-YFP inclusions when the HEK cells were treated with the aggregated solution of 7 μ M α S in the absence and presence of SK-129 at an equimolar ratio. The relative intensity of Proteostat dye-stained aggregates of α S_{A53T}-YFP inclusions (**m**) and relative viability (**n**) of HEK cells treated with the aggregated solution of 7 μ M α S in the absence and presence of SK-129 at an equimolar ratio. Statistical significance, one-way analysis of variance (ANOVA) with Tukey's multiple comparison test. * $p < 0.05$, ** $p < 0.01$, *** $p < 0.001$.

Comment:

Since SK-129 competes with aSyn for the membrane, it would be good to know if SK-129 does also not inhibit the native function of aSyn, resulting in a loss of function. The authors should at least discuss this.

Address to the comment:

We really appreciate reviewer's comment about discussing the function of aSyn in the presence of SK-129 under lipid membrane conditions. It has been suggested that one of the native functions of α S is facilitated by its interaction with the lipid membrane and the complete displacement of aSyn from the lipid membrane could alter the native function of α S. Based on our experimental data in the manuscript and the comparison of our data with literature reports under similar conditions^{42,43}, we have discussed the effect of SK-129 on the membrane catalyzed aggregation and the native function of aSyn in the main manuscript. A separate manuscript will be communicated in the near future, which will demonstrate the effect of SK-129 on the membrane catalyzed aggregation of aSyn and on the function of aSyn in more detail. That manuscript will also demonstrate the role of various binding sites of SK-129 (on aSyn) on the native function of aSyn.

The discussion has been added to the manuscript and it reads as:

"Collectively, our data from CD and NMR suggest that the conformation of α S remains in the α -helical state in the presence of SK-129 for the whole time course of the experiments. The data also indicate that α S was not completely displaced from lipid membranes in the presence of SK-129 and the lipid catalyzed aggregation of α S was wholly inhibited by SK-129. Our CD and NMR data suggest that the inhibition of the membrane-catalyzed aggregation of α S might be a consequence of the competition of α S between lipid membranes and SK-129. A recent study has suggested that one of the main therapeutic strategies could be the inhibition of α S aggregation on lipid membranes without completely displacing α S from lipid membranes^{42,43}. The native function of α S is partly facilitated by its interaction

with lipid membranes and the complete displacement of α S from lipid membranes could be detrimental to its function and might promote neuropathology^{42,43}. A natural product, Squalamine, was able to inhibit the membrane-potentiated α S aggregation and rescued cytotoxicity by completely displacing α S from lipid membranes^{42,43}. However, SK-129 was able to inhibit the aggregation of α S without completely displacing it from lipid membranes, which is evidenced by the intact α -helical conformation of α S in the presence of SK-129 under lipid membrane conditions.

We also employed HSQC 2D-NMR to gain molecular insights into the mode of action of SK-129 on lipid membrane catalyzed aggregation of α S and the overall effect of SK-129 on the membrane bound- α S complex. SK-129 (140 μ M) was added to the complex of ¹⁵N α S:LUVs (70 μ M: 875 μ M) and the intensity changes of the amide peaks from this NMR (Supplementary Fig. 16) were compared with the amide peaks of the NMR from the α S:SK-129 complex (Supplementary Fig. 11,13) and the α S:LUVs complex (Supplementary Fig. 12). The addition of SK-129 to the α S-LUVs complex leads to the disappearance of various amide peaks in the NMR spectrum (Supplementary Fig. 16). If SK-129 was able to completely displace α S from the LUVs, this NMR spectrum should have been similar to the NMR spectrum of the α S-SK-129 complex (Supplementary Fig. 11). However, the NMR spectrum was not similar to that of the α S-SK-129 complex or the α S-LUVs complex. The NMR spectrum was a combination of the NMRs of α S-SK-129 and α S-LUVs complexes, which suggests an interchange of α S between SK-129 and LUVs (Supplementary Fig. 16). The NMR also suggest that SK-129 did not completely displaced α S from the LUVs. Our study demonstrates that SK-129 was able to inhibit membrane catalyzed α S aggregation without completely displacing α S from lipid membranes. The study suggesting that SK-129 is likely not interfering with the native function of α S, which is partly facilitated by the interaction of α S with the lipid membranes.

SK-129 was able to inhibit the aggregation (de novo and lipid membrane conditions) and it was very effective in rescuing PD phenotypes in various cellular and *in vivo* PD models. If SK-129 was only able to inhibit α S aggregation and has interfered with the native function of α S, we would not have observed a significant rescue of toxic functions in various biological systems from PD phenotypes. Collectively, our data suggest that SK-129 potently inhibits α S aggregation without significantly interfering with the native function of α S”.

Comment:

The amplification cycles with the kinetics of seeding of the PMCA assay from patient and control brain is not shown (or I could not find it). It is not possible to judge the kinetics and the quality of the PMCA without the actual data.

Address to the comment:

We agree with reviewer’s comment that is it not possible to judge the kinetics and the quality of the PMCA without the actual data. We have now conducted the aggregation kinetics of α S templated by seeds from Parkinson’s Disease (PD) brain and the control brain. The seeding kinetics of α S monomer was performed in the absence and presence of seeds from the control brain and the PD fibrils in the absence and presence of SK-129 at an equimolar ratio. The results have been included in the supporting information as Supplementary Fig. 21. Also, the writeup has been included in the main text as follows.

Supplementary Fig. 21. a, The aggregation profile of 70 μ M α S (black) catalyzed in the presence of the control brain sample (pink) and the PD brain sample (fibers, blue) in (1 \times PBS buffer). The aggregation profile of 70 μ M α S catalyzed by PD brain samples in the presence of SK-129 at an equimolar ratio (orange). **b**, Comparison of t_{50} 's (The time required to reach 50% fluorescence intensity of ThT dye) for the aggregation profiles indicated in 'a'. The aggregation kinetics of various conditions were conducted three times and the reported t_{50} for various conditions is an average of three separate experiments. The reported error bars are the s.d.'s for three separate experiments.

Writeup in the main text:

“Both samples from the control brain and the PD brain were assessed for their ability to seed and accelerate the aggregation of α S monomer. No noticeable change was observed in the t_{50} or the total ThT intensity for α S aggregation in the presence of the control brain sample. The t_{50} for α S aggregation was 67.4 ± 12.2 and 62.1 ± 3.4 in the absence and presence of the control brain sample, respectively (Supplementary Fig. 21a,b). In marked contrast, both t_{50} (22.7 ± 5.4) and ThT intensity (~ 4 fold) for α S aggregation were significantly enhanced in the presence of the PD brain sample (Supplementary Fig. 21a,b). Clearly, the α S seeds from the PD brain sample template and significantly accelerate α S aggregation via seed catalyzed mechanism.”

Comment:

What is exactly shown in figure 4d and 4e? Are these TEM images of assemblies after extraction (and how were they then extracted and purified) or after amplification?

Address to the comment:

We apologize for the lack of clarification of Fig. 4d and Fig. 4e. The TEM images shown in Fig. 4d and 4e are the fibers extracted from the control and PD brain samples, respectively. The amplification process of α S aggregation led to enhanced fibers (4 fold higher ThT signal) and much shorter t_{50} in the presence of PD brain sample, in contrast to the control brain sample (Supplementary Fig. 21). The process of the extraction and purification of the α S seeds from the PD brain is explained in the method section of the manuscript. We followed an already published protocol to extract and purify the seeds from the PD brain sample³⁴.

We have added the information about the TEM images in the main manuscript and now it reads:

“The Lewy bodies (LBs)-like structural features in the substantia nigra of the PD brain were confirmed using immunostaining (bluish/black, black arrows) (Fig. 4c). We did not observe any LBs-like structural

features in the control brain (Fig. 4b). The aggregates of α S were extracted from the PD brain using a published protocol³⁴. The α S aggregates were extracted and confirmed from the PD brain using TEM (Fig. 4e) and western blot (Fig. 4f). In marked contrast, no aggregates of α S were detected after extraction from the control brain as confirmed by TEM (Fig. 4d)".

Comment:

The discussion is rather limited, and I would like to read more about what can be expected for distribution, BBB permeability, stability in vivo and its usefulness as a drug? Will there be any follow up experiments examining these effects in vivo?

Address to the comment:

We agree with reviewer's comment that the discussion in the manuscript is limited. We have now expanded the discussion in the manuscript. In the discussion, we have included the expectations of SK-129 as a potential drug/lead therapeutics for the treatment of PD. Also, we have discussed the follow up experiments to examine the antagonist effect of SK-129 on the PD phenotypes in an *in vivo* model.

The write up for the discussion is below, which is also incorporated in the main manuscript in the 'Discussion' section:

"The modulation of α S aggregation by affibodies^{58,59} and molecular chaperones⁶⁰ could be an attractive therapeutic intervention for PD; however, proteins/peptides are limited with poor cell permeability and poor enzymatic and conformational stability in biological milieus. Similarly, SK-129 has demonstrated chaperone/affibody-like ability to manipulate α S aggregation and it was able to efficiently rescue PD phenotypes in both *in vitro* and in vivo PD models. The intracellular antagonist activity of SK-129 in various PD models suggests that it possess good pharmaceutical properties, including good cell permeability, enzymatic stability, and structure stability because all of these properties are required for its activity against intracellular α S aggregation. The OQs have been previously shown to maintain potent antagonist activity against their therapeutic targets and demonstrated good cell permeability, structure stability, and enzymatic stability in the biological milieu. A PD mouse model-based study is underway to further assess the pharmaceutical properties and the antagonist activity of SK-129 against PD phenotypes. We are using a well-established mouse model of PD (α SA_{53T} transgenic line M83)^{61,62}. This mouse model has been studied extensively because it mimics the PD pathologies^{61,62}. Using this model, we will be able to assess the pharmacokinetics and pharmacodynamics properties of SK-129. Also, the mouse model study will be used to assess the ability of SK-129 to cross the blood brain barrier. The mouse model study will be used to assess the antagonist activity of SK-129 against PD phenotypes. In addition, the PD mouse model will be treated with PD fibrils in the absence and presence of SK-129. The study will be used to assess the effect of SK-129 on the spreading and propagation of PD phenotypes facilitated by PD fibrils. We are optimistic that the pharmaceutical properties and the antagonist activity of SK-129 can be further optimized without sacrificing its overall conformation. The side chain functionalities of SK-129 scaffold can be conveniently modified synthetically without disturbing its overall conformation for further optimization of activity, which is often challenging with proteins/peptides. Additionally, the C- (COOMe of SK-129) and N-terminus (-NO₂ of SK-129) of SK-129 can also be modified to tune various pharmaceutical properties, including solubility and permeability etc. These manipulations in the chemical structure of SK-129 will not significantly alter its antagonist activity as we have seen with a fluorescent analog of SK-129 (SK-129_F), which has almost similar affinity to SK-129 against α S".

FINAL COMMENT:

We would like to thank all the reviewers for their comments and inputs to further strengthen this manuscript. We believe that we have carefully addressed all concerns raised by the reviewers. All the changes in the main manuscript and supporting information are highlighted in “Yellow”.

Reviewers' Comments:

Reviewer #1:

Remarks to the Author:

The authors addressed most of my comments. I would suggest that the authors include the aS/DOPS titration as the authors performed the experiments rather than just stating that it "yielded a $K_d > 5 \mu\text{M}$ (data not shown)".

Reviewer #3:

None